# Cross-lineage protection by human antibodies binding the influenza B hemagglutinin

Yi Liu[1], Hyon-Xhi Tan[1], Marios Koutsakos[1], Sinthujan Jegaskanda[1], Robyn Esterbauer[1], Danielle Tilmanis[2], Malet Aban[2], Katherine Kedzierska[1], Aeron C. Hurt[1,2], Stephen J. Kent[1,3,4] & Adam K. Wheatley [1]

Influenza B viruses (IBV) drive a significant proportion of influenza-related hospitalisations yet are understudied compared to influenza A. Current vaccines target the head of the viral hemagglutinin (HA) which undergoes rapid mutation, significantly reducing vaccine effectiveness. Improved vaccines to control IBV are needed. Here we developed novel IBV HA probes to interrogate humoral responses to IBV in humans. A significant proportion of IBV HA-specific B cells recognise both B/Victoria/2/87-like and B/Yamagata/16/88-like lineages in a distinct pattern of cross-reactivity. Monoclonal antibodies (mAbs) were reconstituted from IBV HA-specific B cells, including mAbs providing broad protection in murine models of lethal IBV infection. Protection was mediated by neutralising antibodies targeting the receptor binding domain, or via Fc-mediated functions of non-neutralising antibodies binding alternative epitopes including the IBV HA stem. This work defines antigenic cross-recognition between IBV lineages and provides guidance for the rational design of improved IBV vaccines for broad and durable protection.

[1] Department of Microbiology and Immunology, The Peter Doherty Institute for Infection and Immunity, University of Melbourne, Melbourne, VIC 3000, Australia. [2] World Health Organization (WHO) Collaborating Centre for Reference and Research on Influenza, The Peter Doherty Institute for Infection and Immunity, Melbourne, VIC 3000, Australia. [3] Melbourne Sexual Health Centre and Department of Infectious Diseases, Alfred Hospital and Central Clinical School, Monash University, Melbourne, VIC 3004, Australia. [4] ARC Centre for Excellence in Convergent Bio-Nano Science and Technology, University of Melbourne, Parkville, VIC 3010, Australia. Correspondence and requests for materials should be addressed to S.J.K. (email: skent@unimelb.edu.au) or to A.K.W. (email: a.wheatley@unimelb.edu.au)

nfluenza viruses are highly infectious respiratory pathogens that inflict annual epidemics and periodic global pandemics, a significant cause of mortality and morbidity. Two types of influenza viruses co-circulate in human populations, influenza A viruses (IAV) and influenza B viruses (IBV). Both IAV and IBV are formulated into current seasonal influenza vaccines. IBV historically accounts for ~20% of influenza-related hospitalisations in any given year[1], and can dominate influenza seasons as occurred in 2017/2018 in Europe[2]. The clinical severity of IBV is equivalent to that of IAV[3], with children and young adults most susceptible to IBV. Unlike IAV, which naturally resides within aquatic bird populations, IBV infection is almost exclusively restricted to humans, with only rare reports of infection in seals[4]. Mice and ferrets can however be experimentally infected with IBV, providing useful small animal models of human IBV infection and disease[5]. Owing to the limited host range and a markedly slower mutation rate than IAV[6], circulating IBV strains exhibit more limited antigenic diversity than H1N1 and H3N2 IAV[7]. Nevertheless, since the first reports in 1940s IBV has gradually diverged into two distinct lineages—B/Victoria/2/87-like and B/Yamagata/16/88-like[8] (referred to as B/Victoria and B/Yamagata lineages from here on), which are further divided into antigenic clades[9].

An increased awareness of the burden of IBV, coupled with growing antigenic complexity, prompted the recent deployment of quadrivalent influenza vaccines, incorporating components derived from both circulating IBV lineages in order to broaden coverage. Quadrivalent vaccines limit the likelihood of reduced vaccine effectiveness due to IBV lineage mismatch compared to superseded trivalent vaccines, which is thought to have significantly contributed to preventable deaths[10]. Nonetheless, vaccine effectiveness of seasonal influenza vaccines against IBV can often remain low even when antigenic match with circulating strains is good[11,12]. Furthermore, protective serological titres elicited by current seasonal vaccines wane in immunised adults[13]. There remains a need for improved next-generation IBV vaccines to provide resilient and durable protection against a broad array of IBV strains.

While the extensive characterisation of human immune recognition of IAV spans many decades, the principal antigenic targets of IBV, including the degree of cross-recognition of the different lineages by B cells and antibodies, remain poorly defined. Cross-reactive humoral responses between IBV lineages were described as early as 1991[14], with more recent analysis of antibody repertoires after influenza vaccination suggesting a large fraction of antibodies elicited by primary infection[15] or seasonal immunisation[16] are cross-reactive against both lineages of IBV. Selected human monoclonal antibodies (mAbs) have been isolated and can bind and neutralise diverse IBV isolates from both lineages[17–20], highlighting that cross-protective humoral immunity is feasible. These mAbs bind conserved epitopes localised within either the globular head, or the stem domain of hemagglutinin (HA), and can protect mice from challenge with diverse strains of IBV. Protection likely depends, at least in part, upon the elicitation of antibody-dependent cellular cytotoxicity (ADCC)[17]. In addition to HA-specific antibodies, a recent study suggests antibody binding the viral neuraminidase might similarly allow cross-protection against both IBV lineages[21]. Unlike the large number of mAbs characterised for IAV[22], human immune recognition of IBV remains understudied. A greater understanding where and how human B cells and antibodies bind to IBV, the capacity of each site to mediate protection, and the extent of antigenic cross-recognition between IBV lineages is critical to guide efforts to develop improved influenza vaccines.

Using a flow cytometry-based approach previously described for IAV[23,24], we interrogated the IBV-specific memory B cell response following seasonal tri- and quadrivalent vaccination of adult human volunteers. Both classical antibodies inhibiting hemagglutination, as well as novel broadly cross-reactive antibodies binding both IBV lineages were isolated from vaccinated subjects and demonstrated antiviral activity in vitro and in vivo. Furthermore, we demonstrate that protection mediated by non-neutralising antibodies was dependent upon interaction with cellular Fc receptors. The accurate definition of cross-reactive human anti-IBV antibody specificities will guide the design of novel IBV vaccines for broader protection.

## Results

**Development of B cell probes for influenza B hemagglutinin.** Recombinant HA analogues derived from IAV[24] have proven utility for characterising B cells responses to influenza A vaccination using flow cytometry[23,25–27]. We designed analogous constructs to enable expression of soluble trimeric IBV HA ectodomains from B/Phuket/3073/2013 (B-PH13). We generated a wild-type (WT) and a T139G variant, selected to potentially disrupt hydrogen bonds within the receptor binding pocket[28]. With the HA proteins expressed and using flow cytometry (gating in Supplementary Fig. 1), co-staining human memory B cells from influenza immune adults with both WT and T139G HA probes from B-PH13 revealed largely overlapping patterns of staining (Supplementary Fig. 2), which suggested limited antigenic changes between WT and the T139G mutant. However, while significant non-specific association with human B cells was previously described for IAV probes[24], this did not appear the case for B-PH13, and non-mutated HA was deemed suitable for use as B cell probes. The antigenic conservation of the B-PH13 HA probe as compared to the WT virus was further confirmed by staining B cells in mice experimentally infected mice with B/Phuket/3073/2013 (Supplementary Fig. 3). Analogous WT probes for two additional B/Yamagata-lineage HAs (B/Brisbane/60/2008 (B-BR08), B/Yamagata/16/1988), three B/Victoria-lineage HAs (B/Victoria/2/1987, B/Malaysia/2506/2004, B/Florida/4/2006) and the historical B/Lee/1940 strain were also developed.

The utility of IBV probes for tracking IBV-specific B cell responsiveness was first assessed in a cohort of healthy adults ($N = 30$) receiving 2015 trivalent Southern Hemisphere inactivated influenza vaccine (IIV3), including a B-PH13 component[23,29]. Vaccine-elicited expansion of B-PH13+ class-switched memory B cells (CD19+, IgD−, IgG+) was clearly evident between baseline (d0) and d28 post-immunisation (representative staining in Fig. 1a; gating in Supplementary Fig. 1), with a significant twofold rise in observed frequencies from a median 0.18% (IQR: 0.14–0.22) to 0.35% (IQR: 0.22–0.44) (Fig. 1b). The rise in memory B cell expansion was of a similar magnitude to levels we previously reported[23] for vaccine components H3N2 A/Switzerland/9715293/2013 (H3-SW13) and H1N1 A/California/07/2009 (H1-CA09) (Fig. 1c). We also observed increased proportions of B-PH13+ B cells adopting a characteristic activated phenotype (CD27+CD21−) following IIV3 immunisation (Fig. 1d). Moreover, a weak but significant correlation ($r = 0.40$, $p = 0.034$, Spearman's rank-order test) was observed between the magnitude of IIV3-induced expansion in B-PH13+ memory B cell frequencies and the increase in strain-matched serum hemagglutination inhibition assay (HIA) titres (Fig. 1e), confirming our previous observations in IAV studies linking B cell and serologic responsiveness to IIV3[23]. Together, these data highlight memory B cell dynamics following IIV3 are broadly similar to IAV and confirm the utility of IBV HA-derived flow cytometric probes for interrogating IBV-specific B cell immunity in response to vaccination or infection.

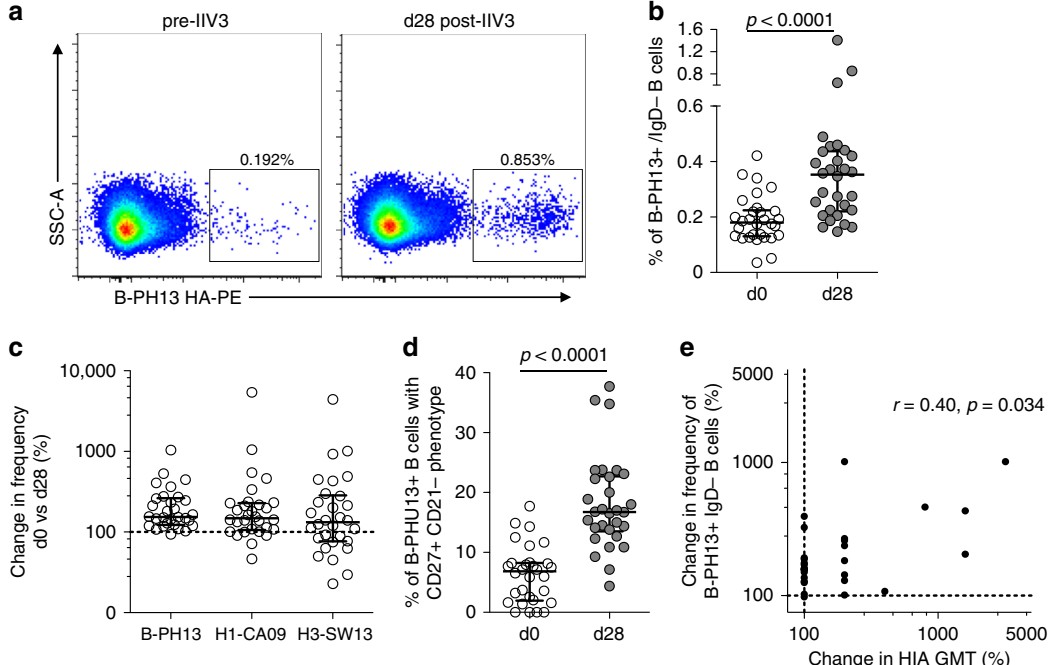

**Fig. 1** Memory B cell responses to B/Phuket/3073/2013 (B-PH13) HA following IIV3 immunisation. **a** Cryopreserved PBMC samples were stained with a panel of monoclonal antibodies and recombinant HA probes allowing frequencies of class-switched memory B cells (CD19+CD20+IgD−) B cells specific for B-PH13 to be assessed. **b** Frequencies of B-PH13 + memory B cells prior to and following IIV3 vaccination (N = 30). **c** Degree of B-PH13+ memory B cell expansion (%) compared with responses to influenza A vaccine components A/Switzerland/9715293/2013 (H3-SW13) and A/California/07/2009 (H1-CA09). **d** The proportion of B-PH13+ memory B cells with an activated memory (CD27+CD21−) phenotype before and after IIV3 immunisation. **e** Correlation between vaccine-elicited expansion of B-PH13+ memory B cell frequencies and changes (%) in HIA geometric mean titre (GMT). Data are shown as median ± interquartile range. Populations were compared using Mann–Whitney U tests and p values are denoted. Correlations were assessed using Spearman's rank-order test and r and p values are denoted

**Memory B cell responses to IIV3 and IIV4.** To better combat IBV viral diversity, many health authorities recommend the quadrivalent (IIV4) over trivalent (IIV3) for seasonal immunisation programmes. The ability of IIV4 to simultaneously promote serological responses to both B/Victoria/2/87-like and B/Yamagata/16/88-like components has been confirmed[30]. However, the dynamics of B/Yamagata and B/Victoria-lineage cross-reactivity at the level of the B cell remains unclear. We therefore compared lineage-specific and cross-reactive B cell responses by flow cytometry following either IIV3 or IIV4 vaccines containing B-PH13 (B/Yamagata component only) or B-PH13 and B-BR08 (combined B/Yamagata and B/Victoria components), respectively. Class-switched memory B cells from subjects receiving IIV3 or IIV4 were co-stained with B-PH13 and B-BR08 probes as before. Four populations of HA-binding memory B cells could be delineated (representative plot in Fig. 2a): B-PH13-specific, B-BR08-specific and two cross-reactive populations, one which bound both probes with roughly equivalent intensity ("cross-reactive - equivalent" or (CR-E)) and one which showed increased relative binding to the B/Yamagata probe (B-PH13) ("cross-reactive – B/Yamagata preference" or (CR-Y)). CR-Y staining was difficult to clearly differentiate from the B-PH13+ population of cells. However, the CR-Y was identifiable in many subjects examined both at baseline and following IIV (Supplementary Fig. 4). Moreover, the specificity of the CR-Y staining pattern was not artefactual as confirmed by swapping the probe SA conjugates on matched samples (Supplementary Fig. 5). Thus, the CR-Y population appears to represent sub-population of B-PH13 HA-specific B cells with weak cross-reactive recognition of B-BR08. No comparable populations of B cells with preferential recognition of B/Victoria were observed. The extent to which preferential binding of cross-reactive memory B cells to the B/

Yamagata probes is generalisable to all IBV strains is not clear. However, this observation was recapitulated in representative donors using probes derived from historical IBV strains B/Florida/4/2006 (B/Yamagata-lineage) and B/Malaysia/2506/2004 (B/Victoria-lineage) (Supplementary Fig. 6).

The responsiveness of each memory B cell population following IIV3 and IIV4 immunisation was compared. B-PH13 + memory B cells were present at a median level of 0.17% (IQR: 0.13–0.22) at baseline in both cohorts and underwent significant expansion (Fig. 2b) in response to both IIV3 and IIV4. In contrast, B-BR08+ memory B cell responses were present at a lower level of 0.04% (IQR: 0.03–0.06) in both cohorts and expanded only in response to IIV4 (Fig. 2c). Interestingly, both CR-Y and CR-E populations expanded in response to IIV3 and IIV4 (Fig. 2d, e). Therefore, seasonal vaccines elicited expansion of both memory B cells specific for the respective IBV component antigens and cross-reactive B cell populations, however the magnitude of CR-Y expansion was significantly greater following IIV4 (Fig. 2f). The serological implications of cross-reactive B cell expansion are currently unclear. However, we observed consistently greater expansion in serum endpoint titres against diverse IBV strains in subjects receiving IIV4 compared to subjects receiving IIV3 (Supplementary Fig. 7), suggesting antibody binding both lineages might be elicited as part of a broad anti-IBV polyclonal response.

**Isolation of IBV-specific human monoclonal antibodies.** We next examined the specificity and breadth of monoclonal antibodies recovered from each of the four identified IBV HA-specific B cell populations. As with our previous IAV work, PBMCs from three IIV4 vaccine recipients with marked expansion of

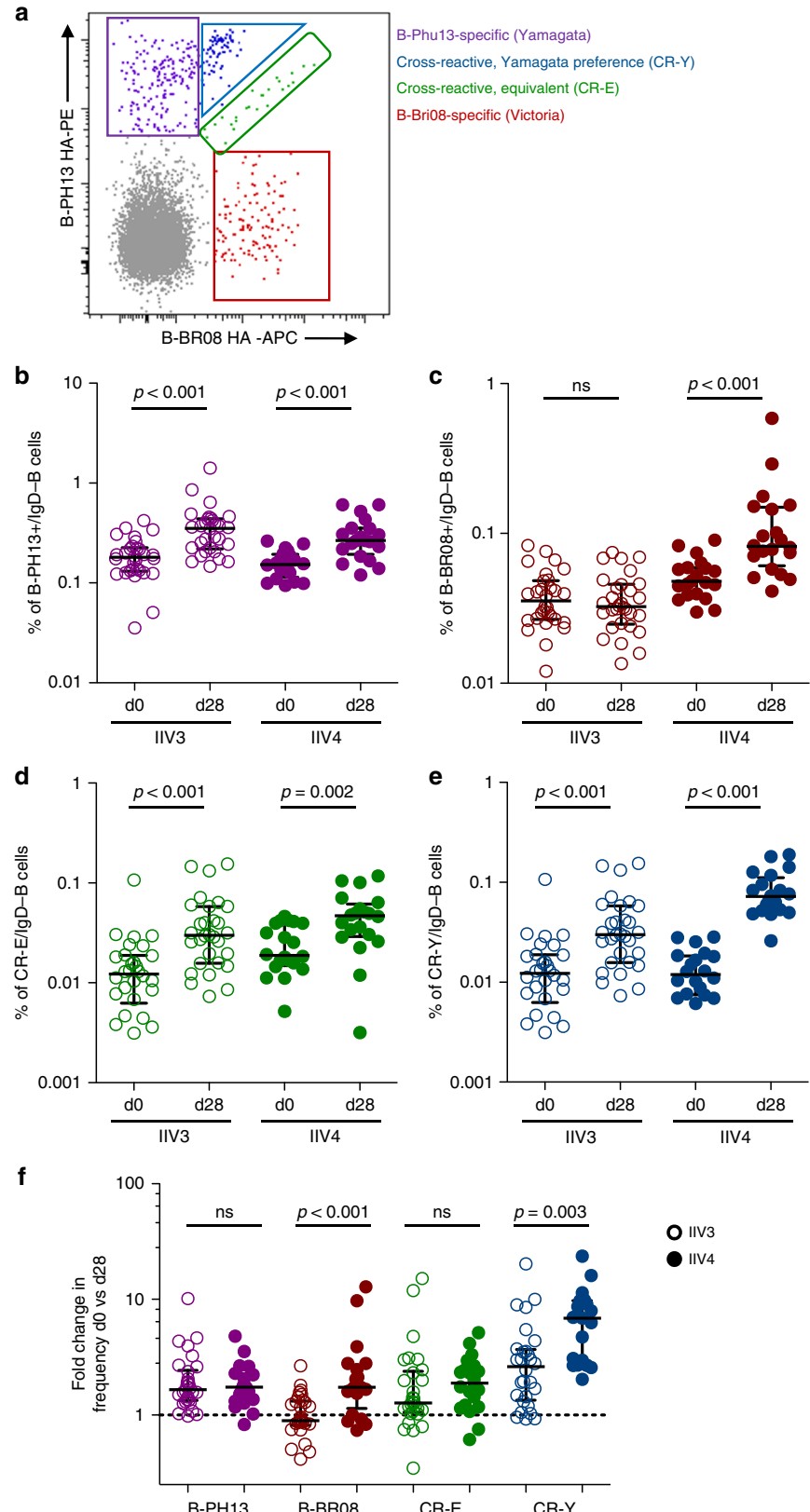

**Fig. 2** Cross-reactive memory B cell responses to IBV HA following IIV3 and IIV4 immunisation. **a** Co-staining class-switched memory B cells with B/Phuket/3073/2013 (B-PH13) and (B/Brisbane/60/2008) B-BR08 probes allowed the delineation of four distinct populations of B cells: B-PH13 (purple) and B-BR08 specific (red), and two cross-reactive populations—one binding each B lineage equivalently (CR-E; green) and one with increased staining intensity for the Yamagata antigen B-PH13 (CR-Y; blue). Frequencies of memory B cells at baseline (d0) and d28 post-immunisation for **b** B-PH13 specific, **c** B-BR08 specific, **d** CR-E and **e** CR-Y populations. **f** Fold change in B cell frequency of each population between d0 and d28. Data are shown as median ± interquartile range. Populations were compared using Mann–Whitney *U* tests and *p* values are denoted. ns not significant

IBV-specific B cells post-immunisation were stained as before, followed by sorting of single B cells and BCR-sequencing as described[24,31]. Out of 672 sorted B cells, 519 productive heavy sequences were recovered, 303 of which also had productive light chain sequences. On the basis of similarities in variable gene usage, complementarity-determining region 3 (CDR-H3) sequence length and CDR-H3 amino acid sequence, we identified 284 unique clonal lineages with a mix of larger clonal expansions and singletons in each population (Supplementary Fig. 8; Supplementary Data 1). A selection of clonally expanded strain-specific or cross-reactive B cell lineages (Supplementary Fig. 9) were chosen for recombinant expression and testing by ELISA for binding to IBV HA. Antibodies isolated from B-PH13-specific ($N = 10$) or B-BR08-specific ($N = 10$) B cell populations displayed HA recognition largely restricted to the sample of three IBV lineage-specific HAs, with the exception of mAb W85–3F06, which was broadly cross-reactive (Fig. 3). The majority of these lineage-specific antibodies demonstrated both HIA and neutralisation activity in vitro, suggesting binding to canonical epitopes that surround the receptor binding site[32]. However, unlike IAV where HIA+ antibodies are usually highly strain specific (i.e. with minimal cross-reactivity even within the same HA subtype), we found anti-IBV mAbs elicited by seasonal vaccines generally recognised all strains tested within a respective IBV antigenic lineage spanning over 20 years of antigenic drift.

Consistent with dual binding of both IBV HA probes by flow cytometry, mAbs isolated from the two cross-reactive B cell populations (CR-Y and CR-E) generally displayed much broader IBV reactivity across both lineages, with greatest breadth observed within the CR-E-derived mAbs. A subset of cross-reactive mAbs could mediate HIA and neutralisation in vitro, suggesting binding of epitopes proximal from the receptor binding domain. Eleven broadly cross-reactive antibodies were identified which bound all 7 IBV HAs tested across the two lineages and the historical B/Lee/1940 strain. As many broadly reactive IAV-specific mAbs bind the HA stem, we generated recombinant IBV stem constructs (Supplementary Figure 10) based upon the designs employed to generate the IAV stem domain[33]. These preliminary recombinant IBV stem proteins appeared misfolded and failed to bind the prototypic stem mAb CR9114, indicating antigenic changes compared to full-length HA (Supplementary Figure 10). However, the IBV stem proteins were bound by three putative broadly cross-reactive IBV stem-specific mAbs: W85–3F06, R95–1E03 and R95–2A08. Stem specificity was further supported based upon binding to full-length HA but not to purified HA1 proteins by ELISA (Fig. 3). Therefore, based on breadth of HA recognition and antiviral activity, we defined three groupings of mAbs with IBV cross-reactivity for in vivo efficacy testing: broad HIA+ lineages (e.g. R95–1E07, R95–1D05, K77–2D09), broad HIA− lineages (e.g. R95–1F04, R95–1C01, R95–1E05) and broad HIA− stem-binding lineages (e.g. W85–3F06, R95–1E03, R95–2A08).

**Protective capacity of human anti-IBV mAbs against IBV.** We next assessed the protective capacity of selected human mAbs in mice experimentally challenged with human IBV strains. Passive infusion of classical HIA+ B/Yamagata-lineage-specific mAbs (K77–1G12, W85–1B01) protected mice from mortality or weight loss following infection with the B/Yamagata-lineage B/Florida/4/2006 virus (Fig. 4a). In contrast, control mice infused with a Victoria-lineage-specific mAb R95–1E12 exhibited no protection. All broadly cross-reactive lineages isolated from CR-E B cells provided some measure of protection against the B/Florida/4/2006 challenge. On the basis of area under the curve analysis of weight maintenance, there was a consistent trend for mAbs which

displayed HIA activity in vitro to be more protective against weight loss than lineages lacking HIA, with the IBV stem-binding mAb R95–1E03 showing the weakest protection.

In mice challenged with B/Victoria-lineage B/Malaysia/2506/2004 virus, we observed three distinct patterns of protection by the panel of mAbs studied. Passive infusion of B/Victoria-lineage-specific HIA+ mAbs provided near complete protection from weight loss (Fig. 4b). Again, we observed superior protection mediated by cross-reactive mAbs with HIA activity, compared to those without HIA activity. The intermediate protection elicited by the three cross-reactive HIA− mAbs was highly consistent. In contrast to the weak protection observed by the stem-specific mAb against the B/Florida/4/2006 challenge, IBV stem-specific mAbs were not protective at all against B/Malaysia/2506/2004, clustering closely to negative controls.

Studies using IAV have established antibody-based protection in the murine challenge model can be mediated via direct neutralisation of virus and/or engagement with host effector cells via Fc receptors (FcR)[34,35], with the ability to engage FcR being highly dependent on epitope location on the viral HA[13,36]. We therefore cloned and re-expressed a selection of IBV-specific mAbs using an Fc backbone with LALA mutations (L234A/L235A) shown to prevent engagement with murine FcRs[37]. Passive infusion of either B/Yamagata-lineage-specific mAbs or the cross-reactive mAb R95–1E07, all of which mediated HIA activity in vitro could still fully protect animals from weight loss in the context of a LALA Fc (Fig. 5). In contrast, cross-reactive mAbs without HIA activity (R95–1C01, R95–1F04) and the stem-specific mAb R95–1E03, lost any protective efficacy in the absence of FcR engagement, suggesting cell-mediated antibody effector functions are critical for protection in vivo, consistent with observations for IAV-specific mAbs[34].

**Mapping cross-protective epitopes on IBV HA.** Pre-dominant antigenic sites of IBV cluster around the receptor binding domain (RBD), while previously resolved epitopes from broadly reactive mAbs bind epitopes located across both the globular head and stem domains of HA (Fig. 6a). When looking at amino acid conservation across the lineages, most divergence is concentrated within RBD proximal regions (Fig. 6b). However, significant areas of pan-IBV conservation exist in the stem, residual esterase domain and in selected areas of the RBD.

In order to better understand the molecular basis of broad IBV recognition by antibody, we mapped the epitope specificity of a selection of IBV neutralising mAbs through generating escape viruses by serial passaging of IBV in the presence of mAbs. Viral supernatants were recovered and the sequence of the full-length HA gene determined[38]. To confirm the selection of viral escape mutants, monoclonal isolates were recovered by plaque purification and similarly sequenced. We first determined the sensitivity of WT and mutant viruses to mAb-directed neutralisation (Supplementary Figure 11). Any loss of recognition of the viral HA from mutant viruses was assessed by flow cytometry using infected cells (Supplementary Figure 12) and by ELISA using recombinant HA proteins (Supplementary Figure 13). Amino acid substitutions conferring antibody resistance were mapped onto resolved IBV HA structures. B/Victoria-lineage-specific mAbs were mapped using B/Brisbane/60/2008 virus. Escape mutants displayed a T214P mutation (R95–1D08, K77–2E02), which potentially abrogated the N-linked glycan motif at N212, and was generally coupled with a three-residue deletion from K177 to D179 (Fig. 6c). While the T214P mutation alone was sufficient to drive escape from antibody-mediated neutralisation, the three-residue deletion conferred additional loss of HA binding by mAbs CR8033 and K77–2E02. B/Yamagata-lineage-

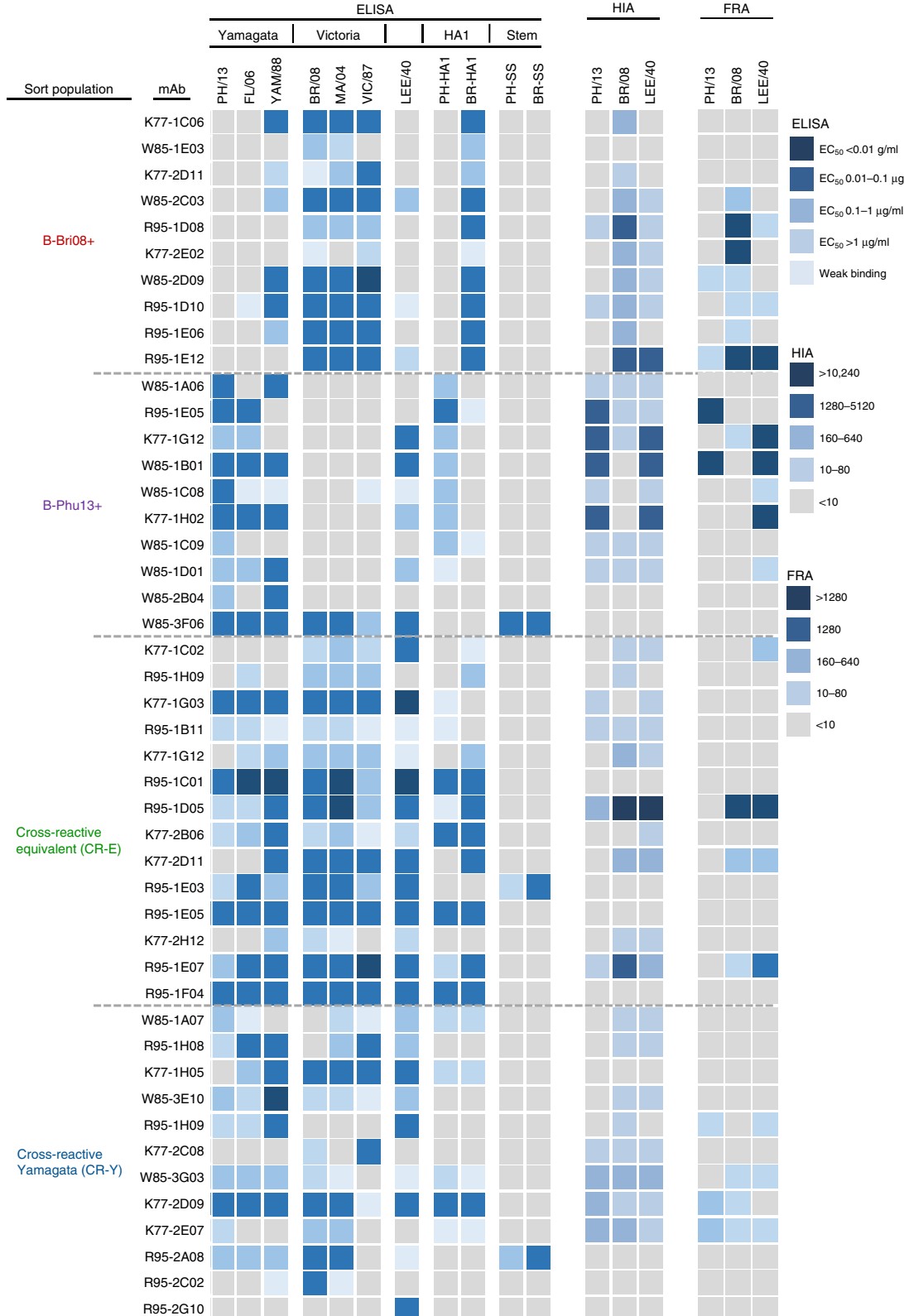

**Fig. 3** Specificity and antiviral activity of recovered human monoclonal antibodies. mAbs were assessed for binding to diverse IBV HA, HA1 and HA stem proteins by ELISA, with the strength of binding calculated as the concentration yielding half-maximal binding (EC50) and denoted by the indicated shading. HIA was assessed as the ability of mAb to inhibit hemagglutination of red blood cells. The dilution of a 1 mg/mL stock with positive HIA signal is denoted by the indicated shading. The neutralisation of IBV viruses was assessed by focus reduction assay (FRA). The dilution of a 1 mg/mL stock, which neutralised 50% of the virus control is denoted by the indicated shading

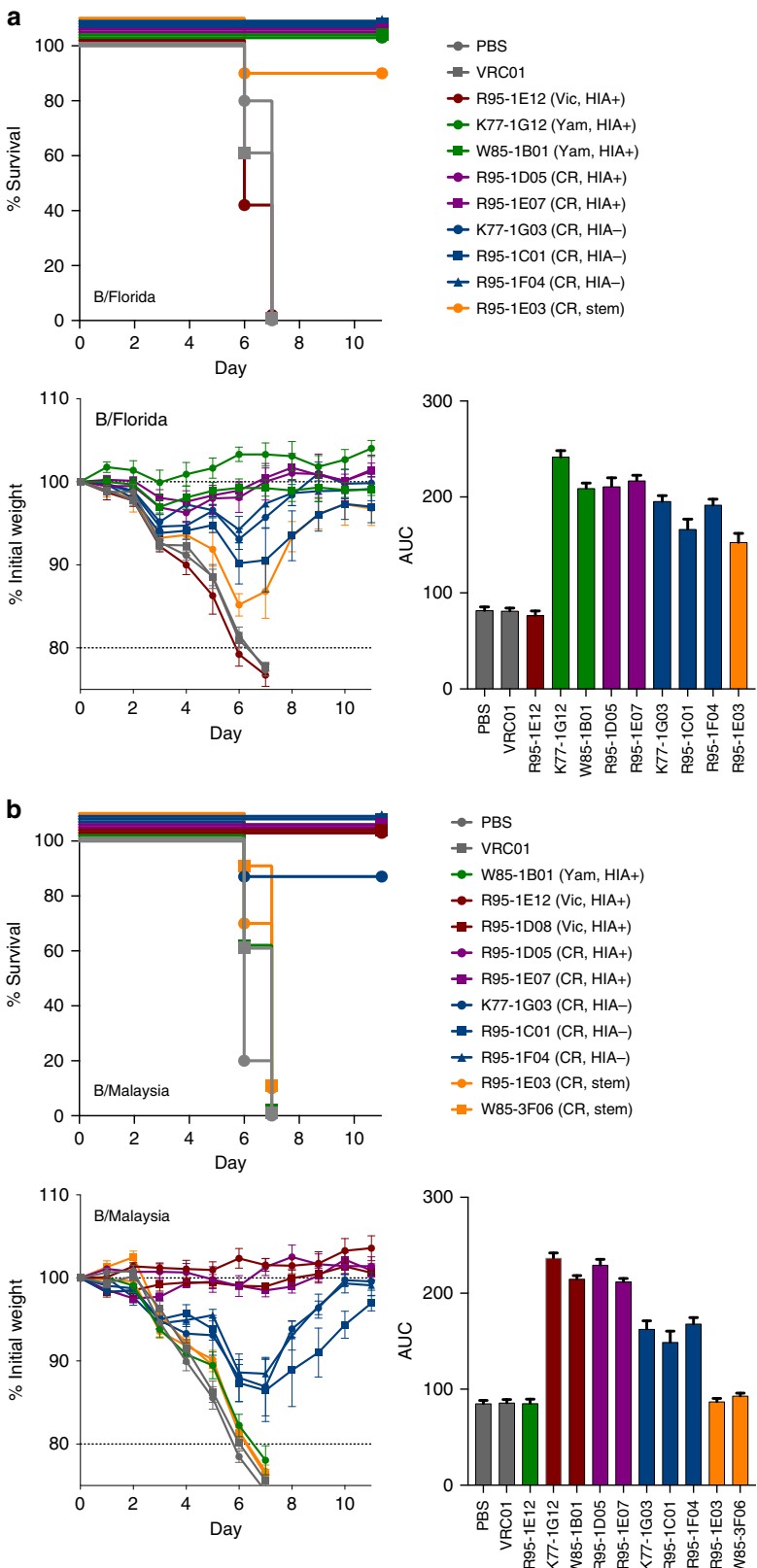

**Fig. 4** Passive protection of mAbs in mice experimentally challenged with influenza mice (N = 5 per group) were infused with 5 mg/kg of anti-IBV mAbs or controls (PBS, VRC01) 24 h prior to intranasal challenge with **a** 50 TCID$_{50}$ B/Florida/4/2006 or **b** 2000 TCID$_{50}$ B/Malaysia/2506/2004. Survival curves (NB: lines are non-overlapping for clarity) and animal weights were monitored over an 11 day period, with animals killed if weight loss exceeded 20% of starting weight. Data is shown as mean ± SEM. Controls (grey), B/Yamagata-specific (red), B/Victoria-specific (green), cross-reactive HIA-positive (magenta), cross-reactive HIA-negative (blue) and cross-reactive stem mAbs (yellow) are indicated. The relative protection of each mAb assessed using area under the curve (AUC) analysis and plotted as mean ± SEM. Yam: B/Yamagata-specific lineage, Vic: B/Victoria-specific lineage, CR: cross-reactive, HIA+: showed HIA activity in vitro, HIA−: no HIA activity in vitro, stem: bound epitopes within the HA stem

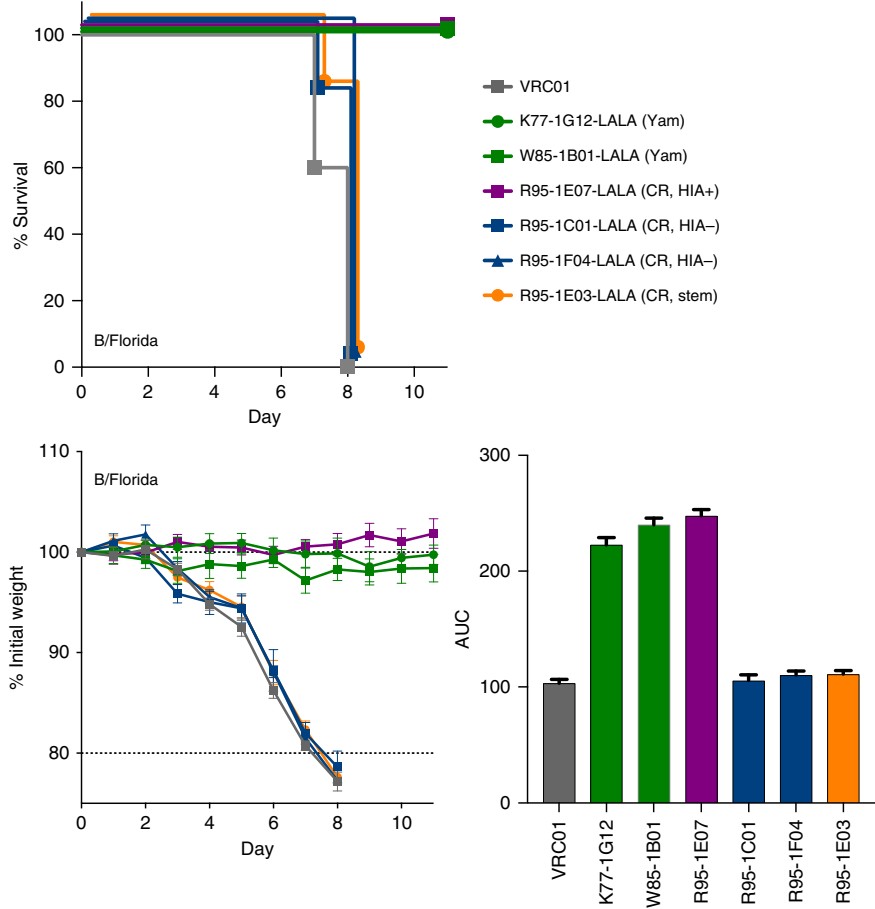

**Fig. 5** Protection by IBV stem or HIA− mAbs requires engagement with cellular Fc receptors mice (N = 5 per group) were infused with 5 mg/kg of anti-IBV mAbs expressed on a human IgG1 Fc with incorporated L234A/L235A (LALA) mutations. Twenty-four hours later, mice were challenged with 50 TCID$_{50}$ B/Florida/4/2006. Animal weights were monitored over an 11 day period, with animals killed if weight loss exceeded 20% of starting weight. Survival curves and weight loss are plotted as mean ± SEM, and the relative protection of each mAb assessed using area under the curve (AUC) analysis and plotted as mean ± SEM. Controls (grey), B/Yamagata-specific (red), B/Victoria-specific (green), cross-reactive HIA-positive (magenta), cross-reactive HIA-negative (blue) and cross-reactive stem mAbs (yellow) are indicated. Yam: Yamagata-specific lineage, CR: cross-reactive, HIA+: showed HIA activity in vitro, HIA−: no HIA activity in vitro, stem: bound epitopes within the HA stem

specific mAbs were mapped using B/Phuket/3073/2013, and convergent pathways of viral escape were observed with substitutions clustered at residues G156 (W85–1B01, K77–1G12), T214I (R95–1E05, K77–1G12) and D212N (K77–1G12, R95–1E05) (Fig. 6d). These mutations are proximal to the receptor binding pocket and localised within the 150-loop and 190-helix structures, respectively. A G156R substitution alone or in combination with D212N mediated neutralisation escape and a partial to complete loss of HA binding by W85–1B01 and K77–1G12. G156E/T214I mutations drove neutralisation escape and a complete loss of HA recognition by K77–1G12, as well as W85–1B01 and CR8033. In terms of cross-reactive mAbs, escape mutations were identified within the viral supernatants upon culturing with R95–1D05, with G156R substitutions in B/Phuket/ 3073/2013 and N212S glycan loss in B/Brisbane/60/2008 (Fig. 6e). However, plaques were not recoverable from the viral supernatant preventing further HA-binding analysis. Escape mutations were similarly generated for a control antibody CR8033, which elicited a T214P substitution and K177 to D179 deletion in B/Brisbane/ 60/2008. Overall, our data indicate the glycine at position 156, and the presence or absence of glycan at position 212, constitute key pathways of escape against both neutralising strain-specific and cross-reactive mAbs. Escape variants could not be generated for IBV stem-specific mAbs, nor for those lacking HIA activity

in vitro, with further epitope definition likely requiring X-ray crystallographic approaches or similar.

## Discussion

IBV infections make up a significant proportion of the global influenza burden[1], with health authorities recommending simultaneous immunisation with both IBV lineages to combat increasing IBV diversity and minimise the chances of seasonal vaccine mismatch. Using a novel flow cytometric approach, we provide a detailed characterisation of the memory B cell responses to IBV HA following immunisation with seasonal tri- and quadrivalent vaccines. We found that each vaccine efficiently drove the expansion of memory B cells binding vaccine component HAs, either B-BR08 or B-PH13. In contrast to these human data, ferret antisera raised by IBV infection displays typically narrow, near strain-specific serological responses to HA with limited inter-lineage cross-reactivity[7], consistent with antigenic changes brought about by drift. In humans, repeated lifetime exposures to both co-circulating lineages renders antibody and B cell recognition of IBV HA decidedly more complex. In line with this, we found a subset of human mAbs initially thought likely to be strain-specific based on single HA probe binding often demonstrated relatively broad cross-lineage HA recognition in vitro and protected from heterologous challenge in vivo,

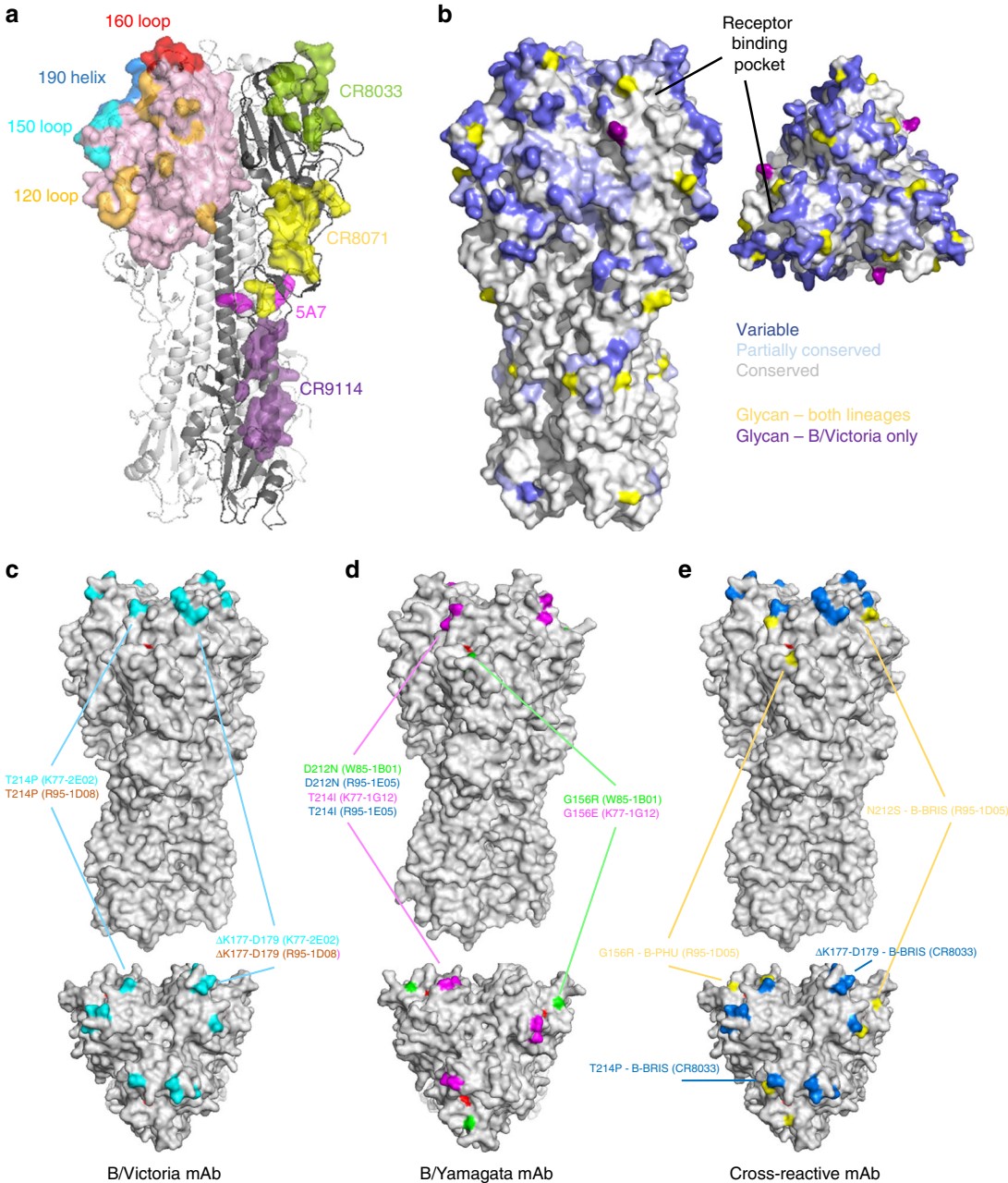

**Fig. 6** Areas of the IBV HA allowing cross-reactive recognition and/or escape from human mAbs. **a** Surface representation illustration of canonical epitopes localised within the receptor binding domain (190-helix, 160-loop, 150-loop, 120-loop) shown on the HA trimer of B/Brisbane/60/2008 (PDB:4FQM). The resolved epitopes of broadly cross-reactive mAbs CR8033, CR8071, 5A7 and CR9114 are illustrated on the adjacent protomer. **b** Amino acid conservation across IBV strains (N = 1000 per lineage) is illustrated on the B/Brisbane/60/2008 HA trimer with shading indicating areas of low (<90%), partial (>90%) and near complete conservation. Sites of N-linked glycosylation are indicated for both lineages (yellow) and for B/Victoria only (purple). **c** Escape mutations elicited by B/Victoria-specific mAbs are mapped onto the HA of B/Brisbane/60/2008 (PDB:4FQM). F95 within the receptor binding pocket indicated in red. **d** Escape mutations elicited by B/Yamagata-specific mAbs are mapped onto the HA of B/Yamanashi/166/1998 (PDB:4M40). **e** Escape mutations elicited by cross-reactive mAbs are indicated on B/Brisbane

despite binding to canonical epitopes such as the 150-loop and 190-helix, which are subject to antigenic drift in human populations[7].

The elicitation of IBV cross-reactive serological responses to seasonal vaccines has been previously reported[14,39–41]. When assessed directly at the level of the B cell, two interesting and reproducible populations of cross-reactive B cells (CR-Y and CR-E) were identified, with recovered mAbs confirming broad IBV recognition by ELISA. Pan-IBV recognition was concentrated within mAbs derived from the CR-E population, with 64% (9/14)

of mAbs binding all 7 IBV HA tested compared to 16% (2/12) of the CR-Y-derived mAbs. Nevertheless, it remains possible that the CR-E and CR-Y populations might derive from continuous spectrum of cross-reactivity and additional mAb isolation and characterisation is warranted. While administration of IIV3 was sufficient to drive expansion of both CR-Y and CR-E memory B cell populations, expansion of CR-Y was significantly enhanced following IIV4 immunisation. It remains to be seen if a differential ability to drive expansion of CR-Y B cells contributes to any potential increased serological protection from IIV4 vaccines. A

limitation of our study is the restricted number of mAbs and the heterogenous patterns of IBV HA binding and neutralisation. However, in general we find seasonal influenza vaccines drive expansion of two pre-dominant populations of memory B cells; one binding neutralising epitopes proximal to the receptor binding site shared within each respective IBV lineage, and a second population of highly cross-reactive B cells binding both lineages and expressing antibodies that were generally non-neutralising. Although many human mAbs displayed a degree of protection in mice, the protective benefit of cross-reactive non-neutralising antibodies in human populations remains an open question. Nevertheless, IIV4 drove expanded serological reactivity against a broad range of IBV strains and simultaneously expands B cells expressing HIA+ antibody with pan-B/Yamagata or pan-B/Victoria reactivity, suggesting IIV4 has an improved capacity to combat antigenic drift in either lineage while removing the risk of lineage mismatch relative to IIV3.

Interestingly, reports from immunised children[39] and in murine models[42] found that prior priming with B/Yamagata antigens followed by immunisation with B/Victoria induces strong serological recall of B/Yamagata, but only low B/Victoria antibody responses. The converse is not the case, suggesting a degree of immunologic dominance of B/Yamagata HA. Elements of our study are consistent with these observations, such as (a) lower baseline frequencies of B/Victoria-specific memory B cells in adults compared with B/Yamagata-specific memory B cells, (b) a cross-reactive B cell population (CR-Y) that shows preferential binding to B/Yamagata HA by flow cytometry and (c) the absence of cross-reactive B cells that show the converse preferential binding to B/Victoria HA. Further resolution of the precise epitopes recognised by IBV cross-reactive B cells and antibodies may help elucidate mechanisms that underpin any differential capacity of antigens from the two IBV lineages to recall cross-reactive immunity.

Several human antibodies with broad recognition of both IBV lineages have been isolated and are of interest as potential diagnostic, therapeutic and prophylactic agents for IBV infection. Broad IBV recognition and HIA activity was first demonstrated by the isolation of CR8033, which binds and neutralises both IBV lineages and has HIA activity against B/Yamagata strains[20]. Subsequently, the humanised murine antibody 12G6 was shown to potently neutralise and elicit HIA activity against both IBV lineages[19], and binds a similar epitope to CR8033 overlapping the receptor binding domain. We identified at least four mAbs (R95–1D05, K77–2D11, R95–1E07, K77–2D09) with analogous breadth and HIA activity in vitro. Viral escape mutants generated with R95–1D05 and CR8033 both entailed glycan removal at N212, suggesting binding to highly similar epitopes. Interestingly, conserved pathways of escape were observed for both lineage-specific and cross-reactive mAbs that mediated HIA. For B/Yamagata, this was focused at two regions: substitutions in the 150-loop at the glycine residue 156 (G156K, G156E, G156R), and substitutions at residues 212 and 214 within the 190 helix. Analogous G156R and G156E mutations were reported for cross-reactive mAb C12G6[19] and double substitutions at D212 and T214 previously reported for B/Yamagata-specific human mAbs 3A2 and 10C4[18]. For B/Victoria, we commonly observed removal of glycan at N212 via N212S (R95–1D05) or T214P (R95–1D08, K77–2E02, CR8033) mutations, generally coupled with deletion of K177 to D179 (R95–1D08, K77–2E02, CR8033) in the 160-loop. These HA modifications lie immediately proximal to residues identified in B/Brisbane/60/2008 essential for binding by cross-reactive mAb C12G6[19]. Common pathways of escape for both lineage-specific and cross-reactive mAbs may complicate the design of IBV HA immunogens for preferential induction of pan-IBV immunity. In addition, HIA+ cross-reactive antibody specificities like CR8033 might be susceptible to a loss of efficacy in the face of antigenic drift, potentially driven by strain-specific or lineage-specific humoral responses at the population level.

Broad IBV recognition and broad prophylactic and/or therapeutic protection against IBV infection in vivo can be mediated at alternative epitopes. In particular, within the vestigial esterase domain at the base of the HA, as seen with CR8071[20] and 46B8[17], or localised in the IBV stem domain, as seen with CR9114[20]. While we have identified antibodies with properties suggestive of both CR8071-like (W85–1A07, W85–3E10) and HA stem-reactive specificities (W85–3F06, R95–1E03, R95–2A08), the absence of neutralisation activity in vitro means accurate epitope localisation via crystallographic or electron microscopy approaches is required.

The ready detection of broadly cross-reactive B cells in humans based on our study presented here is supportive of efforts to broaden vaccine protection. Moreover, a recent study using phage display to derive mAbs from a seasonal vaccine recipient also identified multiple cross-reactive antibody lineages including those with intra-Yamagata, inter-lineage and even CR9114-like specificities[43]. However, the degree to which B cells targeting the different epitope clusters are recruited into protective serological responses remains unclear, although some initial studies suggest cross-reactive serum antibody may be significant[16]. In terms of protective capacity, we observed a clear hierarchy following passive infusion studies whereby cross-reactive antibodies capable of mediating HIA showed the greatest protective effect in vivo. This is consistent with the reported activity of mAb 12G6, which demonstrated greater prophylactic and therapeutic protection against experimental IBV challenge than antibodies binding distal to the receptor-binding domain[44]. Next in that order were mAbs that conferred intermediate protection, characterised by broad IBV recognition but no HIA activity, in a manner akin to CR8071. Consistent with reports for IAV[34,35], protection mediated by such mAbs was highly dependent upon engagement with cellular Fc receptors. Finally, we found mAbs binding the IBV stem domain failed to neutralise in vitro and provided the weakest protection against experimental challenge. The fact that mAb CR9114 can neutralise IAV but not IBV strains[20] suggests the IBV stem may exist in a conformation comparatively resistant to antibody-mediated neutralisation. This is supported by the poor epitope accessibility and low relative neutralising potency of the IBV stem-proximal mAb 5A7[18]. In contrast, a recent report detailed the isolation of multiple human IBV stem-binding antibodies that can both neutralise in vitro and provide potent cross-lineage protection in mice[45]. In human populations, serum antibodies binding the IBV stem are widely prevalent, with titres increasing with age[46] or following IBV infection[15]. However, while the IBV HA stem is highly conserved and can be targeted in mice to protective effect[47], the utility of the IBV stem as a human vaccine target remains to be clarified.

In summary, we demonstrate broadly cross-reactive memory B cells are common in humans, express both neutralising and non-neutralising immunoglobulins and are efficiently recruited into the humoral response elicited by seasonal immunisation. Further, IBV cross-reactive mAbs demonstrate protective potential in vivo and may constitute a substrate amenable to selective targeting by improved vaccine designs. Accurate definition of conserved sites of vulnerability on the IBV HA, in particular, clarifying the molecular basis of narrow versus broad IBV HA recognition, will greatly inform rational vaccine design efforts for eliciting broad and durable protective immunity against IBV.

## Methods

**Ethics statement**. The study protocols were approved by both the Alfred Hospital Ethics Committee (# 432/14), and the University of Melbourne Human Research

Ethics Committee (# 1443420) and all associated procedures were carried out in accordance with the approved guidelines. All participants provided written informed consent in accordance with the Declaration of Helsinki. Animal studies and related experimental procedures were approved by the University of Melbourne Animal Ethics Committee (#1714193).

**IIV3 and IIV4 clinical samples**. The 2015 IIV3 immunisation trial is fully described elsewhere[29] and registered as NCT02632578. Briefly, 30 healthy Australian adults (mean age 40.4 years) were vaccinated with the 2015 IIV3 (bioCSL Fluvax®) containing 15 μg of hemagglutinin from A/California/7/2009-like (pdmH1N1), A/Switzerland/9715293/2013 (H3N2)-like and B/Phuket/3073/2013-like strains. Blood samples were taken at baseline (d0) and d28 post-immunisation and sera, plasma and PBMCs were cryopreserved prior to use. For the 2016 IIV4 trial, 20 healthy adults were administered Sanofi FluQuadri® vaccine containing A/California/07/2009-like virus (pdmH1N1), A/Hong Kong/4801/2014-like virus (H3N2), B/Phuket/3073/2013-like virus and B/Brisbane/60/2008-like virus components. Sera, plasma and PBMCs were collected and cryopreserved from d0 and day 28. Participant information is summarised in Supplementary Figure 14.

**HA proteins and probes**. Recombinant HA proteins for use as flow cytometry probes derived for IAV A/California/7/2009 and A/Switzerland/9715293/2013 strains were previously described[23,24]. Analogous probes were prepared for influenza B encompassing the HA ectodomain C-terminally fused to the trimeric foldon of T4 fibritin, a biotinylatable AviTag sequence GLNDIFEAQKIEWHE, and a hexa-histidine affinity tag (sequences in Supplementary Figure 15). Briefly, synthetic genes encompassing the HA ectodomain from B/Phuket/3073/2013, B/Brisbane/60/2008, B/Yamagata/16/1988, B/Victoria/2/1987, B/Malaysia/2506/2004, B/Florida/4/2006 and B/Lee/1940 and variants containing the T129G mutation to limit sialic acid binding were synthesised (GeneArt) and cloned into mammalian expression vectors. Stabilised IBV stem protein expression plasmids were constructed based upon designs established to work for IAV[33]. Recombinant HA and HA stem proteins were expressed by transient transfection of Expi293F (Life Technologies, A14527) suspension cultures and purified by polyhistidine-tag affinity chromatography and gel filtration. For use as flow cytometric probes, HA proteins were biotinylated using BirA (Avidity) and stored at −80 °C. Prior to use, biotinylated HA proteins were labelled by the sequential addition of streptavidin (SA) conjugated to PE, APC, Alexa488 (ThermoFisher) or BV421 (BD) and stored at 4 °C.

**Flow cytometry**. HA-specific B cells were identified within cryopreserved human PBMC by co-staining with HA probes conjugated to SA-PE, SA-APC, SA-BV421 or SA-Ax488. Monoclonal antibodies for surface staining included: CD19-ECD (Beckman Coulter IM2708U, J3–119, 1:150), CD20 Alexa700 (BD 560631 2H7, 1:300), IgM-BUV395 (BD 563903, G20–127, 1:75), CD21-BUV737 (BD 564437, B-ly4, 1:150), IgD-Cy7PE (BD 561314 IA6–2, 1:500), IgG-BV786 (BD 564230, G18–145, 1:150), CD14-BV510 (Biolegend 301841, M5E2, 1:300), CD3-BV510 (Biolegend 344828, OKT3, 1:600), CD8a-BV510 (Biolegend RPA-T8, 1:1500), CD16-BV510 (Biolegend 302047, 3G8, 1:500), CD10-BV510 (Biolegend 312220, HI10a, 1:750) and CD27-BV605 (Biolegend 302829, O323, 1:100). Background B cells interacting with SA were excluded by staining with SA-BV510 (BD 563261). Cell viability was assessed using Aqua Live/Dead amine-reactive dye (ThermoFisher). For mouse samples, splenocytes or lymph node suspensions were stained with the following panel: B220-BUV737 (BD 564449, RA3–6B2; 1:300), IgD-BUV395 (BD 564274, 11–26 c.2a, 1:300), CD45-Cy7APC (BD 557659, 30-F11, 1:300), GL7-Alexa488 (Biolegend 144612,GL7, 1:300), CD38-Cy7PE (Biolegend 90, 1:750), SA-BV786 (BD 563858, 1:300), CD3-BV785 (Biolegend 100355, 145–2C11, 1:750) and F4/80-BV785 (Biolegend 123141, BM, 1:150). Samples were collected using a BD Fortessa configured to detect 18 fluorochromes and analysis was performed using FlowJo software version 9.5.2 (TreeStar).

**Sequencing, cloning and expression of B cell immunoglobulins**. The sequencing and cloning of BCRs from single B cells and the expression of recombinant mAbs was performed as previously described[24,31,48]. Productive, recombined heavy chain (V-D-J) and light chain (V-J) immunoglobulin sequences were synthesised (Genscript), cloned into expression plasmids and transfected into Expi293F cells using ExpiFectamine (Invitrogen). Recombinant monoclonal antibodies were purified from culture supernatants using sepharose Protein-A or G (Pierce). BCR sequences have been deposited in Genbank accession MK311355 - MK312159.

**ELISA**. Antibody binding to IBV HA, HA1 and stabilised stem proteins was tested by ELISA. 96-well Immunosorp plates (ThermoFisher) were coated overnight at 4 °C with 2 μg/mL recombinant IBV HA expressed in Expi293 cells or sourced commercially (Sino Biological). After blocking with 1% fetal calf serum (FCS) in PBS, duplicate wells of IBV-specific monoclonal antibodies at different dilutions (starting at 10 μg/mL, four times serial dilutions) or human sera (1:100, four times serial dilutions) were added and incubated for one hour at room temperature. Plates were washed prior to incubation with 1:30,000 dilution of HRP-conjugated anti-human IgG (KPL) for 1 h at room temperature. Plates were washed and developed using 3,3′,5,5′-Tetramethylbenzidine (TMB) substrate (Sigma) and read

at 450 nm. HA-binding intensity was calculated as the antibody concentration giving half-maximal signal (EC$_{50}$) using a fitted curve (4 parameter log regression). For serum samples, endpoint titres were using a fitted curve (4 parameter log regression) and a cutoff of two times background.

**Passive infusion of IBV-specific monoclonal antibodies**. B/Florida/4/2006 and B/Malaysia/2506/2004 viruses were grown in embryonated eggs and infectious titres were determined by 50% tissue culture infective dose (TCID$_{50}$) viral assays. Challenge stocks were titrated in mice and assessed for pathogenicity (Supplementary Figure 16). Groups of five C57BL/6 female mice at 8–12 weeks of age were used to examine prophylactic potential of mAbs generated. Mice were administered mAbs intraperitoneally at a dose of 5 mg/kg body weight. Twenty-four hours after passive infusion of mAbs, mice were anaesthetised by isoflurane inhalation and intranasally challenged with 50 μL of B/Florida/4/2006 and B/Malaysia/2506/2004 viruses at doses of 50 and 2000 TCID$_{50}$, respectively. Animals were monitored for weight loss and signs of infection for 14 days, and killed if a loss of more than 20% of their pre-infection weight occurs.

**Hemagglutination inhibition assay (HIA)**. HIA activity of recombinant mAbs was assessed using 1% turkey erythrocytes in a standardised assay as previously described[49]. Briefly, mAbs were diluted to 100 μg/mL in PBS prior to incubation with ether treated[50] influenza viruses from strains B/Phuket/3073/2013, B/Brisbane/60/2008 and B/Lee/1940. HIA titres are reported as the reciprocal of the highest dilution where hemagglutination was completely inhibited.

**Focus reduction assays (FRA)**. Neutralisation activity of recombinant mAbs against B/Phuket/3073/2013, B/Brisbane/60/2008 and B/Lee/1940 was examined using focus reduction assays as previously described[51]. The neutralisation titre is expressed as the reciprocal of the highest dilution of a 1 mg/mL mAb stock at which virus infection is inhibited by ≥50%.

**Viral escape assay**. The generation of IBV escape mutants was based upon previously described protocols[38]. Briefly, 24-well plates were seeded with $2.5 \times 10^5$ Madin Darby Canine Kidney (MDCK) cells (ATCC CCL-34) per well to form confluent monolayers. The next day, serial dilutions of recombinant mAbs were incubated with B/Phuket/3073/2013 and B/Brisbane/60/2008 for one hour at 37 °C in Flu-media (Dulbecco's Modified Eagle's Medium (DMEM) with 0.8% bovine serum albumin (BSA), 1% penicillin/streptomycin and 0.1% L-1-Tosylamide-2-phenylethyl chloromethyl ketone (TPCK)-treated trypsin), before adding the virus-antibody mixture to MDCK cells. After 2 to 3 days culture media supernatants from wells with visible cytopathic effect were collected and used to infect a fresh monolayer of MDCK cells in the presence of increasing concentrations of mAb. After serial passaging, culture supernatants were harvested, viral RNA was extracted and the IBV HA gene sequenced. Putative mutant viruses were identified based upon sequence comparison to similarly passaged media-only or irrelevant mAb (anti-HIV VRC01) controls. Where mutants were identified, single virus isolates were recovered by plaque purification using standard techniques. Single plaques were rescued, expanded in MDCK cells, and mutant viruses within culture supernatants sequenced and TCID$_{50}$ determined as before.

**mAb neutralisation and HA-binding assays**. The neutralisation activity of recombinant mAbs was examined using a modified microneutralisation assay[52]. MDCK cells were seeded in 96-well plates at $1.5 \times 10^5$ per well. The next day, serial dilutions of recombinant mAbs were incubated in Flu-media with 100TCID$_{50}$ of WT or mutant B/Phuket/3073/2013 and B/Brisbane/60/2008 viruses for one hour at 37ºC, before addition to MDCK cells. After 18–24 h, supernatants were removed, cells were fixed and cellular cytopathicity was visualised by ELISA using mouse anti-influenza B nucleoprotein (1:1000;Abcam) primary and goat anti-mouse HRP-conjugated secondary antibodies. Plates were developed using TMB substrate and read at 450 nm. The concentration of mAb preventing 50% infectivity (IC$_{50}$) was calculated.

The ability of recombinant mAbs to bind cell-surface HA on infected cells was examined by flow cytometry. MDCK cells were seeded into 6-well plates and infected with ~10,000 TCID$_{50}$ WT or mutant B/Phuket/3073/2013 and B/Brisbane/60/2008 viruses and incubated at 37 °C for 18–24 h. Cells were resuspended by manual scraping and infectivity confirmed by staining with mouse anti-influenza B nucleoprotein (1:1000; Abcam) and goat anti-mouse Alexa647 (1:5000; ThermoFisher). The binding of human anti-IBV mAbs (5 μg/mL) or an anti-HIV negative control (VRC01) to surface expressed HA was detected using goat anti-human Alexa647 (1:5000; ThermoFisher).

**Sequence analysis of viral isolates**. To characterise conservation of the IBV HA, a cross-section of 2000 B/Yamagata and B/Victoria-lineage viral sequences spanning 1988–2018 were exported from the EpiFlu database [gisaid.org], HA protein sequences aligned using Geneious 11.1.3 (Biomatters) and weighted conservation scores accounting for year of isolation determined at each residue position. Amino acid conservation was visualised using Pymol.

**Statistical analyses**. Data is generally presented as median ± interquartile range. Statistical significance was assessed by Mann–Whitney *U* tests. Correlations were analysed using Spearman's rank-order tests. Curve fitting was performed using 4 parameter logistic regression. All statistical analyses were performed using Prism (GraphPad).

**Reporting summary**. Further information on experimental design is available in the Nature Research Reporting Summary linked to this article.

## Data availability

All data generated or analysed during this study are included in the published article and its supplementary information files.

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

## Acknowledgements
We thank all study participants and the Melbourne Cytometry Platform (MBC node) for expert flow cytometry assistance. This work was supported by NHMRC programme grant ID1052979 (SJK), NHMRC project grant ID1129099 (AKW) and Proof-of-Concept Initiative funding co-provided by the University of Melbourne and Sanofi Pasteur. YL and MK are supported by a Melbourne International Research Scholarship and Melbourne International Fee Remission Scholarship. KK was supported by a NHMRC Senior Research Fellowship Level B. World Health Organization (WHO) Collaborating Centre for Reference Research on Influenza is supported by the Australian Government Department of Health.

## Author contributions
Y.L., H.-X.T., S.J.K. and A.K.W. designed the study. Y.L., H.-X.T., M.K., R.E. and S.J. performed experiments. M.K. and K.K. provided viruses and samples. D.T., M.A. and A. C.H. performed serological analyses and plaque purifications. Y.L., H.-X.T., S.J.K. and A. K.W. wrote the manuscript. All authors read and revised the manuscript.

## Additional information

**Competing interests:** This study was funded in part via a Proof-of-Concept Initiative awarded to A.K.W. and co-funded by Sanofi Pasteur, which is the manufacturer of influenza vaccines (FluQuadri) received by some, but not all, of the participants of this study. The remaining authors declare no other competing interests.

