## [Peer Review File · Nature Communications]

Reviewers' comments:

Reviewer #1 (Remarks to the Author):

In their manuscript Liu et al describe the analysis of the memory B-cell response against influenza B virus HA after influenza virus vaccination. This is a very nice paper that draws attention to the understudied influenza B virus. However, there are several issues that need the authors' attention.

Major points

1) The influenza B stalk construct is not described well. Please include the sequence and data about its folding and recognition by mAbs (e.g. CR9114) needs to be added.

2) The authors find that anti-stalk mAbs did protect against the a Yamagata challenge virus but not against a Victoria-lineage challenge virus. This is surprising given that described anti-stalk mAbs (e.g. CR9114) protect well against both lineages. The authors do not specify the challenge dose in terms of LD50. Differences in LD50ies used might cause this phenomenon. LD50ies should be performed and that data should be provided.

3) There are several issues with the escape mutant generation. What were these viruses compare to? The proper negative controls would be viruses passaged alongside in the presence of irrelevant antibody. Which of the many mutations that the authors found were actually caused by adaptation to cell culture? Also, the authors did not plaque purify the escape cultures to get to monoclonal viruses. This needs to be done. It would also be good practice to clone out the escaped HA to confirm loss of binding in a clean background. Furthermore, data on loss of binding is in general missing and needs to be included.

4) Since no sequences for the probes are provided it is unclear if they had a trimerization domain. If not, its stalk might be misfolded and then it would not be surprising that no potentially protective anti-stalk mAbs were isolated. Also, the HA probes were produced in mammalian cells which usually leads to the attachment of very large N-linked glycan structures (much bigger than typically found on the virus - see <https://www.ncbi.nlm.nih.gov/pubmed/?term=eichelberger+influenza+glycan>). How might this have skewed the analysis?

5) The monoclonal data does not confirm the presence of a CR-Y subset. The CR-Y mAbs shown in figure 3 do not differ in breadth from the CR-E subset. This needs to be discussed and might be an artifact of the FACS analysis.

6) It is disturbing that mAb W85-3F06 which is identified as stalk binder has HI activity. This is unusual and warrants further confirmation.

7) Survival should be shown in the main manuscript in Figure 4.

8) It would be much better to show the mutations in Figure 6 C, D and E on the full trimer.

Minor points

1) Line 25: 'influenza', not 'Influenza'

2) Line 53: Ferrets are actually not really a good model for IBV.

3) Regarding the lineages: Please use B/Victoria/2/87-like and B/Yamagata/16/88-like throughout the manuscript to avoid mix-up (there is strains from Victoria in the B/Yamagata/16/88-like and vice versa).

4) Line 79 and following: It might be worth mentioning crossprotective influenza B NA mAbs as well.

5) In general, how does the data compare to the mAbs recently published by Hirano et al (<https://www.ncbi.nlm.nih.gov/pmc/articles/PMC5964595/>)?

6) Line 190: Please add a ')' after 'Table 1'.

7) Line 225: 'mortality', not 'lethality'

8) Line 372: Anti-stalk antibody titers in humans for H1, H3 and B have been compared by Nachbagauer et al (<https://www.ncbi.nlm.nih.gov/pmc/articles/PMC4725014/>).

9) Please define abbreviations when first used in the text including: FCS, TMB, TCID50, TPCK, DMEM, BSA, MDCK etc.

10) Line 439: 'labeled', not 'labelled'

11) Line 489: 'Inhibition', not 'inhibition'

12) Line 514: 'Viral Isolates'

13) Line 515: If sequences were selected randomly there will be a heavy skew to recent years (because there is much more sequences). A set # of sequences per year should be used and the analysis should be repeated.

14) Figure 2: Non of the axis labeling is legible

Reviewer #2 (Remarks to the Author):

In the manuscript titled "Cross-lineage protection by vaccine-elicited human antibodies binding the influenza B hemagglutinin", the authors used IBV HA probes in antigen-specific FACS sorting to interrogate humoral responses to IBV in humans. The authors found a significant proportion of IBV HA-specific B cells could recognize both Yamagata and Victoria lineages. A large panel of recovered mAbs were cloned and tested in ELISA, HIA and FRA experiments. This study provides clear conclusions and significant amounts of information and experimental data of IBV-reactive mAbs.

Concerns and suggestions:

1. The labels in Figure 2 are unreadable.
2. The authors generated the IBV stem proteins based on previous work for IAV (Yassine et al. 2015 Nature Medicine). HA stems are notoriously unstable and it has been known that stable trimeric HA stems are difficult to produce. It is possible that, in figure 3, the mAbs showed no binding activities to HA1s and stems are actually stem-directed. To prove the quality of the HA stems used in this study, it is recommended and Important that the authors include a diagram of their stem protein constructs and a SDS-PAGE gel (or size exclusion chromatography) showing the stable trimeric IBV stem proteins in supplementary section, which will add significant value to this manuscript.
3. Subjects information was not clearly described in the manuscript. Were the three IIV4 recipients used for FACS sorting and mAbs isolation K77, W85 and R95? Also, including a demographic table summarizing the ethnic and age background of all the participants in this study in supplementary section would be informative.
4. It is interesting that the subjects (supplementary fig 4.) generated CR-E and CR-Y but no CR-V mB cells after immunization. The baseline mB cells of these subjects are also more Yamagata lineage dominant. It would be interesting to know the influenza exposure history (circulating IBV lineage in childhood, vaccination and previous infection record) of these subjects.
5. In the flow cytometry experiments, the authors observed the immune dominance to Yamagata lineage (13/Phuket) in the subjects. This might be an artifact due to the quality (stability, folding, quaternary structures or accessibility to mB cells) of the HA proteins (13PH and 08BR) used. To exclude this possibility, the authors might test other HA proteins (i.e. 06FL and 04MA, etc.) with a few representative subjects in the flow cytometry experiments. Alternatively, an indirect experiment can be performed to further confirm the observation: The authors might perform ELISA to test the subjects' serum antibody titers to HA proteins from different Yamagata and Victoria strains.
6. The authors stated that "out of 672 sorted B cells, 519 productive heavy sequences were recovered."

Please include the number of clones with both heavy chain and light chain sequences recovered.

7. Data in supplementary table 1. is fragmented and unreadable. Also, IGHV information is missing.

Using an excel file to document these data might be better than using a PDF file.

8. A significant number of IBV reactive Ab clones were identified in this study. It might be worth trying to perform statistical analyses on the germline gene usage of the clones found in this study to identify "molecular signatures" of IBV specific antibodies. (Ref: Avnir et al., 2014 PLoS Pathogens & Joyce et al. 2016 Cell)

Reviewer #3 (Remarks to the Author):

In this article by Liu et al., the authors analyze memory B cell (MBC) responses that are directed against influenza B virus HA glycoprotein. The authors use a flow cytometry-based approach to identify and sort HA-specific memory B cells that are specific to Phuket HA (Yamagata lineage) or Brisbane HA (Victoria lineage). The main time point where most of the analysis was performed was 4 weeks after immunization of healthy young adults with the trivalent (contains Yamagata, n=30) or the quadrivalent (contains Yamagata + Victoria, n=20) inactivated seasonal influenza. The authors elegantly show the distinct MBC populations directed against the Phuket HA vs. the Brisbane HA and those that equally bound to both probes. The authors further dissected the response by generating recombinant human mAbs from sorted single HA+ cells and characterizing the binding and functional capacity of the mAbs, confirming the mono-specificity or cross-reactivity of the sorted cells. Some of the mAbs were then shown to be broadly protective in a lethal murine influenza B virus challenge model. Further, the authors explored the role of the Fc-mediated effector functions in the protective capacity of a subset of the mAbs. Finally, the author generated and sequenced viral escape mutants against a subset of the in vitro neutralizing mAbs.

Overall the manuscript is very well-written with clear figures and the study set up is robust given the size of the cohorts. The flow cytometry-based approach that the authors used is innovative.

Comments:

1) In the title, it is inaccurately stated that the analyzed MBCs/mAb are vaccine elicited. While this could be true for a part of the analyzed response, but given the sorting strategy there is no definitive way to tell how much of the sorted MBCs are actually vaccine-induced vs. pre-existing. This is supported by data shown in Fig. 1D, where only a little over 15% of Phuket HA+ MBCs detected at day 28 after vaccination were CD21lo, a sign of recent activation. To truly and properly study vaccine-elicited responses, the authors should have analyzed the d7 plasmablast responses as described in detail by numerous reports over the last 10 years. Recently generated plasmablasts are short lived in blood so no interference from pre-existing cells would exist. Contrary to the perpetuated myth, plasmablasts can be analyzed from cryo-preserved PBMCs especially using flow cytometry.

- 2) The HA staining looks robust, but there is no negative control. An irrelevant, similarly labeled probe should have been used to be able to accurately tell whether all these HA+ MBCs are truly specific. This is especially true for the pre-vaccination timepoint. Without that, the data in Fig. 1B could turn out to be an overestimate.

- 3) The correlation in Fig. 1E is non-sensical as it has been shown multiple times that the increase in serum Abs is after influenza vaccination correlates with the frequency of the antibody-secreting plasmablasts.

- 4) The authors show the binding of their HA probe to murine GC B cells for no apparent reason.

- 5) There are key questions/findings that the authors could have addressed/highlighted, such as the extent to which B cell responses in humans that are directed against the Yamagata lineage vs. Victoria lineages vs. cross-reactive ones are elicited following seasonal vaccination. Do we actually need a quadrivalent vaccine or the trivalent vaccine is enough? Are the data showing that the majority of the cross-reactive mAbs are non-neutralizing compared to strain-specific ones statistically significant? If so, what are the implications? The authors instead were seemingly side-tracked with a population of HA-binding cells that did not express any mAbs with any particularly special capacities. And even that population, the authors did not provide a meaningful explanation for it.

RESPONSES TO REVIEWERS' COMMENTS

We thank the reviewers for their comments and suggestions, which we believe have significantly strengthened the manuscript. We address each reviewers' comments below in point-by-point form. We appreciate the opportunity to re-submit our study to *Nature Communications*.

Reviewer #1 (Remarks to the Author):

In their manuscript Liu et al describe the analysis of the memory B-cell response against influenza B virus HA after influenza virus vaccination. This is a very nice paper that draws attention to the understudied influenza B virus. However, there are several issues that need the authors' attention.

1) The influenza B stalk construct is not described well. Please include the sequence and data about its folding and recognition by mAbs (e.g. CR9114) needs to be added.

We now include a new Supplementary Figure 10, which includes details of the sequence, expression characteristics and antigenic characterisation of the IBV stem probes used in this study. These IBV stem probes enabled the identification of three putative stem mAbs in the current study, further confirmed by lack of binding to the HA1 proteins. However, these initial constructs do not bind the prototypic stem mAb CR9114 and show heterogenous expression by SDS-PAGE and gel filtration, suggesting there are antigenic differences to full-length HA. This work provides a solid basis on which to further improve IBV stem proteins and immunogens over time as more IBV stem antibodies with defined epitopes are reported and characterised.

We have added the following text to the relevant results section:

“Since many broadly reactive IAV-specific mAbs bind the HA stem, we generated recombinant “stabilised” IBV stem constructs (Supplementary Figure 10) based upon the designs employed to generate the IAV stem domain³³. The recombinant IBV stem proteins failed to bind the prototypic stem mAb CR9114, indicating antigenic changes compared to full length HA (Supplementary Figure 10). However, the IBV stem proteins were bound by three putative broadly cross-reactive IBV stem-specific mAbs - W85-3F06, R95-1E03, R95-2A08. Stem specificity was further supported based upon binding to full-length HA but not to purified HA1 proteins by ELISA (Fig. 3).”

And included Supplementary Figure 10:

A

B/Phuket/3073/2013 stem construct

MKAIIVLLMVVTSNADRICTGITSSNSPHVVKTATQGEVNVTVIPL **G**SGLKLANGTKYR**P**Q**R**ETRGFFGAIAGFLEGGWEGMIA
 GWHGYTSHGAHGVAAADLKSTQEAINKITKNLNSLSELE **G**SGSG**T**DLAELAVLLSN**E**GIINSEDEHLLALERK**L**KKMLG**P**SAV
 DIGNGCFETKHKCNQ**T**CLDR**I**AAGTFNAGEFSLPT**F**DSL**N**IT **G**SG**Y**I**P**E**A**PR**D**G**Q**AY**V**R**K**D**G**EW**L**L**S**T**F**L **G**S**G**L**N**D**I**F**E**A**Q**K**I**E
WH**E**G**H**H**H**H**H**H*

B/Brisbane/60/2008 stem construct

MKAIIVLLMVVTSNADRICTGITSSNSPHVVKTATQGEVNVTVIPL **G**SGLKLANGTKYR**P**Q**R**ETRGFFGAIAGFLEGGWEGMIA
 GWHGYTSHGAHGVAAADLKSTQEAINKITKNLNSLSELE **G**SGSG**T**DLAELAVLLSN**E**GIINSEDEHLLALERK**L**KKMLG**P**SAV
 EI**G**NGCFETKHKCNQ**T**CLDR**I**AAGTFDAGEFSLPT**F**DSL**N**IT **G**SG**Y**I**P**E**A**PR**D**G**Q**AY**V**R**K**D**G**EW**L**L**S**T**F**L **G**S**G**L**N**D**I**F**E**A**Q**K**I**E
WH**E**G**H**H**H**H**H**H*

B

C

D

Supplementary Figure 10 – Design and expression of IBV HA stem proteins

(A) Stabilised IBV stem constructs encompassing relevant sections of the IBV HA ectodomain interspersed with linkers (purple) then C-terminally fused to the trimeric foldon of T4 fibrin (red), AviTag (green) and hexa-histidine affinity tag (blue). (B) SDS-PAGE of expressed recombinant IBV proteins. Lane 1 – marker, lane 2 – 5 μ g B/Brisbane/60/2008 HA trimer, lane 3 – 5 μ g B/Phuket/3073/2013 HA trimer, lane 4 – 5 μ g B/Brisbane/60/2008 HA stem, lane 5 – 5 μ g B/Phuket/3073/2013 HA stem. (C) Gel filtration trace of B/Phuket/3073/2013 HA trimer and B/Phuket/3073/2013 HA stem proteins. (D) Binding of known IBV-specific mAbs to stabilised IBV stem proteins and a rHA control was examined by ELISA.

2) The authors find that anti-stalk mAbs did protect against a Yamagata challenge virus but not against a Victoria-lineage challenge virus. This is surprising given that described anti-stalk mAbs (e.g. CR9114) protect well against both lineages. The authors do not specify the challenge dose in terms of LD50. Differences in LD50ies used might cause this phenomenon.

LD50ies should be performed and that data should be provided.

In line with the ethical guidelines governing animal research in Australia, human care of influenza infected animals requires euthanasia when the animal shows signs of morbidity and actual lethality endpoints are not permissible. However, using a weight loss in excess of 20% or other indications of distress as a surrogate, we calculated “LD50” and we now provide the titrations and approximate MLD₅₀ of the B/Florida and B/Malaysia challenge stocks in Supplementary Figure 16. Notably, the challenge viruses used in the current study are not mouse-adapted, and the dose given $\sim 2 \times$ MLD₅₀ B/Florida and $\sim 2.5 \times$ MLD₅₀ for B/Malaysia are broadly in line with other reports in the literature (for example, $3 \times$ MLD₅₀ Chai et al. 2017, $5 \times$ MLD₅₀ Wohlbold et al. 2017)

The following text was added to the methods:

“Challenge stocks were titrated in mice and assessed for pathogenicity (Supplementary Figure 16).”

And Supplementary Figure 16 included:

Supplementary Figure 16 - Titration of mouse challenge stocks

Weight loss in mice receiving increasing intranasal doses of B/Florida/4/2006 or B/Malaysia/2506/2004 challenge stocks.

3) There are several issues with the escape mutant generation. What were these viruses compare to? The proper negative controls would be viruses passaged alongside in the presence of irrelevant antibody. Which of the many mutations that the authors found were actually caused by adaptation to cell culture?

Negative controls used in this study were wild-type virus similarly passaged in the presence of media alone, or the non-influenza specific mAb VRC01 (anti-HIV). No cell culture adaptations were observed for any of the repeatedly passaged negative control viruses. We have expanded the detail in the Methods to make this clearer with the following addition:

“Putative mutant viruses were identified based upon sequence comparison to similarly passaged media-only or irrelevant mAb (anti-HIV VRC01) controls.”

Also, the authors did not plaque purify the escape cultures to get to monoclonal viruses. This needs to be done. It would also be good practice to clone out the escaped HA to confirm loss of binding in a clean background. Furthermore, data on loss of binding is in general missing and needs to be included.

We have now plaque purified the viral supernatants to obtain monoclonal viruses. In general, monoclonal viruses matched the bulk sequenced escape variants with the exception of R95-1D05, where single plaques could not be recovered for analysis. We have added the following details to the Methods:

“Where mutants were identified, single virus isolates were recovered by plaque purification using standard techniques. Single plaques were rescued, expanded in MDCK cells, and mutant viruses within culture supernatants sequenced and TCID₅₀ determined as before.”

As suggested, the monoclonal viruses were used to assess any loss in antibody-mediated neutralisation activity against the escaped viruses. This data is now incorporated as new Supplementary Figure 11:

mAb	Virus	Mutant	IC50
K77-1G12	B/Phuket/3073/2013	WT	<0.05mg/ml
		G156E, T214I	>100mg/ml
W85-1B01	B/Phuket/3073/2013	WT	<0.05mg/ml
		G156R, D212N	>100mg/ml
R95-1D05	B/Phuket/3073/2013	WT	0.095mg/ml
		G156R	2.47mg/ml
K77-2E02	B/Brisbane/60/2008	WT	0.275mg/ml
		Δ K177-D179, T214P	>100mg/ml
		T214P	>100mg/ml
CR8033	B/Phuket/3073/2013	WT	<0.05mg/ml
	B/Brisbane/60/2008	WT	44mg/ml
		Δ K177-D179, T214P	>100mg/ml

Supplementary Figure 11 – Neutralisation activity of human mAbs against wildtype (WT) and escape mutant IBV

The inhibitory concentration of mAb sufficient to prevent infection of 50% of MDCK tissue culture wells (IC₅₀) was determined for monoclonal wildtype and escape mutant viruses recovered after plaque purification.

We have added the following sections to the Methods:

“mAb neutralisation and HA binding Assays

The neutralisation activity of recombinant mAbs was examined using a modified microneutralisation assay {World Health Organization, 2011 #489}. MDCK cells were seeded in 96-well plates at 1.5×10^5 per well. The next day, serial dilutions of recombinant mAbs were incubated in Flu-media with 100TCID₅₀ of wild-type or mutant B/Phuket/3073/2013 and B/Brisbane/60/2008 viruses for one hour at 37°C, before addition to MDCK cells. After 18-24 hours, supernatants were removed, cells were fixed and cellular cytopathicity was visualised by ELISA using mouse anti-influenza B nucleoprotein (1:1000;Abcam) primary and goat anti-mouse HRP-conjugated secondary antibodies. Plates were developed using TMB substrate and read at 450nm. The concentration of mAb preventing 50% infectivity (IC₅₀) was calculated.

The ability of recombinant mAbs to bind cell-surface HA on infected cells was examined by flow cytometry. MDCK cells were seeded into 6-well plates and infected with ~10000 TCID₅₀ wild-type or mutant B/Phuket/3073/2013 and B/Brisbane/60/2008 viruses and

incubated at 37°C for 18-24 hours. Cells were resuspended by manual scraping and infectivity confirmed by staining with mouse anti-influenza B nucleoprotein (1:1000;Abcam) and goat-anti-mouse Alexa647 (1:5000;Thermofisher). The binding of human anti-IBV mAbs (5ug/ml) or an anti-HIV negative control (VRC01) to surface expressed HA was detected using goat-anti-human Alexa647 (1:5000;Thermofisher).”

As suggested, we also assessed the escape mutants for loss of HA binding. This was done using two approaches. Firstly, MDCK cells were infected with wild-type and escape mutant viruses and the binding of mAbs to the surface localised HA was assessed by flow cytometry. The relevant methods are in the section above and the results presented in Supplementary Figure 12. In addition, we cloned and expressed the three most common HA mutants and assessed mAb binding by ELISA. These data are presented in Supplementary Figure 13:

Supplementary Figure 12 – Binding of mAbs to surface HA following infection of MDCK cells

MDCK cells were infected *in vitro* with wild-type (WT) or viruses with the indicated escape mutations. The binding of human mAbs to HA on the cell surface was assessed by flow cytometry 18 hours post-infection. A major loss of binding relative to wildtype virus is indicated in red shading.

Supplementary Figure 13 - Binding of mAbs to wildtype and mutant HA by ELISA

Recombinant HA ectodomains were expressed using sequences from wild-type (WT) or viruses with the indicated escape mutations. The binding of human mAbs to HA was assessed by ELISA.

The relevant text in the Results has been revised and now reads:

“Viral supernatants were recovered and the sequence of the full-length HA gene determined³⁸. To confirm the selection of viral escape mutants, monoclonal isolates were recovered by plaque purification and similarly sequenced. We first determined the sensitivity of wild-type and mutant viruses to mAb-directed neutralisation (Supplementary Figure 11). Any loss of recognition of the viral HA from mutant viruses was assessed by flow cytometry using infected cells (Supplementary Figure 12) and by ELISA using recombinant HA proteins (Supplementary Figure 13). Amino acid substitutions conferring antibody resistance were mapped onto resolved IBV HA structures. B/Victoria-lineage-specific mAbs were mapped

using B/Brisbane/60/2008 virus. Escape mutants displayed a T214P mutation (R95-1D08, K77-2E02), which potentially abrogated the N-linked glycan motif at N212, and was generally coupled with a three-residue deletion from K177 to D179 (Figure 6C). While the T214P mutation alone was sufficient to drive escape from antibody-mediated neutralisation, the three-residue deletion conferred additional loss of HA binding by mAbs CR8033 and K77-2E02. B/Yamagata-lineage-specific mAbs were mapped using B/Phuket/3073/2013, and convergent pathways of viral escape were observed with substitutions clustered at residues G156 (W85-1B01, K77-1G12), T214I (R95-1E05, K77-1G12) and D212N (K77-1G12, R95-1E05) (Figure 6D). These mutations are proximal to the receptor binding pocket and localised within the 150-loop and 190-helix structures, respectively. A G156R substitution alone or in combination with D212N mediated neutralisation escape and a partial to complete loss of HA binding by W85-1B01 and K77-1G12. G156E/T214I mutations drove neutralisation escape and a complete loss of HA recognition by K77-1G12, as well as W85-1B01 and CR8033. In terms of cross-reactive mAbs, escape mutations were identified within the viral supernatants upon culturing with R95-1D05, with G156R substitutions in B/Phuket/3073/2013 and N212S glycan loss in B/Brisbane/60/2008 (Figure 6E). However, plaques were not recoverable from the viral supernatant preventing further HA binding analysis. Escape mutations were similarly generated for a control antibody CR8033, which elicited a T214P substitution and K177 to D179 deletion in B/Brisbane/60/2008. Overall, our data indicate the glycine at position 156, and the presence or absence of glycan at position 212, constitute key pathways of escape against both neutralising strain-specific and cross-reactive mAbs. Escape variants could not be generated for IBV stem-specific mAbs, nor for those lacking HIA activity in vitro, with further epitope definition likely requiring X-ray crystallographic approaches or similar.”

Overall, the new neutralisation and binding data strengthens our observations that the glycine at position 156 and the glycan at position 212 are key mediators of escape within sites proximal to the receptor binding site for both strain-specific antibodies and cross-reactive mAbs, a finding consistent with other reported IBV escape mutants in the literature.

4) Since no sequences for the probes are provided it is unclear if they had a trimerization domain. If not, its stalk might be misfolded and then it would not be surprising that no potentially protective anti-stalk mAbs were isolated. Also, the HA probes were produced in mammalian cells which usually leads to the attachment of very large N-linked glycan structures (much bigger than typically found on the virus - see <https://www.ncbi.nlm.nih.gov/pubmed/?term=eichelberger+influenza+glycan>). How might this have skewed the analysis?

We have now provided the sequences for rHA probes in Supplementary Figure 15 and the following text has been added to the Methods for clarity:

“Analogous probes were prepared for influenza B encompassing the HA ectodomain C-terminally fused to the trimeric FoldOn of T4 fibrin, a biotinylatable AviTag sequence GLNDIFEAQKIEWHE, and a hexa-histidine affinity tag (sequences in Supplementary Figure 15).”

Notably, a foldon is present in these constructs, and the resultant HA proteins are mostly trimeric based upon post-expression purification via gel filtration (examples can be seen in Supplementary Figure 10). In terms of glycosylation, we acknowledge rHA probes may be

decorated with glycans that occupy a slightly larger volume than those found in egg-produced HA or virions (where NA may remove sialic acid substitutions). This may theoretically affect epitope accessibility on the rHA probes. However, we have not to date found significant differences in mAb binding to rHA probes versus HA upon virions where this has been directly tested. For example, in our earlier work we found 24 of 25 murine mAbs specific for defined epitopes of PR8 bound similarly to rHA probes and virus (Figure S2, Whittle et al., 2014).

5) The monoclonal data does not confirm the presence of a CR-Y subset. The CR-Y mAbs shown in figure 3 do not differ in breadth from the CR-E subset. This needs to be discussed and might be an artifact of the FACS analysis.

We agree that based upon staining using rHA probes, clear differentiation between putative CR-E, CR-Y and the B-PHU populations of B cells is difficult. Indeed, these populations might encompass a continuous spectrum of cross-reactivity. We do see more cross-reactivity (defined as binding all 7 IBV HA tested) concentrated in the CR-E population, suggestive that these populations may be somewhat distinct. Supporting this, the CR-Y population more often contained clonal expansions, while the CR-E populations was generally comprised of singletons (Supplementary Figure 6). However, we have amended the Discussion with the following text to better highlight the uncertainty:

“Pan-IBV recognition was concentrated within mAbs derived from the CR-E population, with 64% (9/14) of mAbs binding all 7 IBV HA tested compared to 16% (2/12) of the CR-Y-derived mAbs. Nevertheless, it remains possible that the CR-E and CR-Y populations might derive from continuous spectrum of cross-reactivity and additional mAb isolation and characterisation is warranted.”

6) It is disturbing that mAb W85-3F06 which is identified as stalk binder has HI activity. This is unusual and warrants further confirmation.

It is certainly unusual for mAbs that bind distally from the receptor binding site to mediate HI activity *in vitro*. It is worth noting that the observed HI activity for the mAb W85-3F06 was very weak (HIA titre 10), and markedly less than that observed for neutralising mAbs binding canonical epitopes in the HA head (generally ranging from 80-5120). Notably, the human mAb 5A7 (Yasugi et al., 2013), which binds an epitope localised to the IBV HA stem/stalk has also been reported to mediate weak HI activity, suggesting any unusual properties of W85-3F06 are not unprecedented.

We have updated the Results text with the following:

“Interestingly, mAb W85-3F06 did mediate some weak HAI activity *in vitro*, similar to the previously reported activity of mAb 5A7¹⁸.”

7) Survival should be shown in the main manuscript in Figure 4.

Survival data have been moved from the Supplementary Figures and incorporated into Figure 4 and the relevant Results text amended.

8) *It would be much better to show the mutations in Figure 6 C, D and E on the full trimer.*

Figures 6C-6E have been updated to illustrate mutations upon the full HA trimer.

Minor points

1) *Line 25: 'influenza', not 'Influenza'*

This has been corrected.

2) *Line 53: Ferrets are actually not really a good model for IBV.*

We merely note in the text that ferrets can be experimentally infected with human IBV strains.

3) *Regarding the lineages: Please use B/Victoria/2/87-like and B/Yamagata/16/88-like throughout the manuscript to avoid mix-up (there is strains from Victoria in the B/Yamagata/16/88-like and vice versa).*

We have amended the description of the lineages in the text of the abstract and the introduction. Further, to maintain optimal readability, we define the abbreviations B/Victoria and B/Yamagata to refer to the antigenic lineages and have made any such references now consistent throughout the manuscript. The relevant section in the Introduction now reads:

“Nevertheless, since the first reports in 1940s IBV has gradually diverged into two distinct lineages - B/Victoria/2/87-like and B/Yamagata/16/88-like ⁸ (referred to as B/Victoria and B/Yamagata lineages from here on), which are further divided into antigenic clades ⁹.”

4) *Line 79 and following: It might be worth mentioning cross-protective influenza B NA mAbs as well.*

The following text has been added:

“In addition to HA-specific antibodies, a recent study suggests antibody binding the viral neuraminidase might similarly allow cross-protection against both IBV lineages ²¹.”

The following reference has been added:

21. Wohlbold TJ, *et al.* Broadly protective murine monoclonal antibodies against influenza B virus target highly conserved neuraminidase epitopes. *Nat Microbiol* **2**, 1415-1424 (2017).

5) *In general, how does the data compare to the mAbs recently published by Hirano et al (<https://www.ncbi.nlm.nih.gov/pmc/articles/PMC5964595/>)?*

The recent study by Hirano et al. used phage display to derive mAbs from a single human vaccine recipient. Like us, they too find examples of pan-lineage reactivity and inter-lineage

cross-reactivity in their reconstituted mAbs, and similarly define CR8071- and CR8033-like specificities. We have updated the discussion to acknowledge this finding with the following text:

“Moreover, a recent study using phage display to derive mAbs from a seasonal vaccine recipient also identified multiple cross-reactive antibody lineages including those with intra-Yamagata, inter-lineage and even CR9114-like specificities⁴³.”

The following reference has been added to the References:

43. Hirano D, *et al.* Three Types of Broadly Reacting Antibodies against Influenza B Viruses Induced by Vaccination with Seasonal Influenza Viruses. *J Immunol Res* **2018**, 7251793 (2018).

6) Line 190: Please add a ')' after 'Table 1'.

This has been corrected.

7) Line 225: 'mortality', not 'lethality'

This has been amended.

8) Line 372: Anti-stalk antibody titers in humans for H1, H3 and B have been compared by Nachbagauer *et al* (<https://www.ncbi.nlm.nih.gov/pmc/articles/PMC4725014/>).

The text in the Discussion has been amended to read:

“In human populations, serum antibodies binding the IBV stem are widely prevalent, with titres increasing with age⁴⁶ or following IBV infection¹⁵. However, while the IBV HA stem is highly conserved and can be targeted in mice to protective effect⁴⁷, the utility of the IBV stem as a human vaccine target remains to be clarified.”

The following reference has been added to the References:

46. Nachbagauer R, Choi A, Izikson R, Cox MM, Palese P, Krammer F. Age Dependence and Isotype Specificity of Influenza Virus Hemagglutinin Stalk-Reactive Antibodies in Humans. *MBio* **7**, e01996-01915 (2016).

9) Please define abbreviations when first used in the text including: FCS, TMB, TCID50, TPCK, DMEM, BSA, MDCK *etc.*

These abbreviations are now spelled out in the text.

10) Line 439: 'labeled', not 'labelled'

This has been corrected.

11) Line 489: 'Inhibition', not 'inhibition'

This has been corrected.

12) Line 514: 'Viral Isolates'

This has been corrected.

13) Line 515: If sequences were selected randomly there will be a heavy skew to recent years (because there is much more sequences). A set # of sequences per year should be used and the analysis should be repeated.

We thank the reviewer for this suggestion, which indeed oversampled recent isolates in the initial analysis. We repeated this analysis with conservation weighted relative to the number of sequences for each year of isolation. However, the resultant picture of inter-lineage IBV conservation is largely unchanged. This may be due to increasing divergence between the two lineages (~92.5% homology in 2018 versus ~95.14% homology in 1988) making most variability concentrated in more recent isolates.

The Methods text has been modified with the following text:

“To characterise conservation of the IBV HA, a cross-section of 2000 B/Yamagata and B/Victoria lineage viral sequences spanning 1988 – 2018 were exported from the EpiFlu database (gisaid.org), HA protein sequences aligned using Geneious 11.1.3 (Biomatters) and weighted conservation scores accounting for each year of isolation determined at each residue position. Amino acid conservation was visualised using Pymol.”

The image in Figure 6B was left unchanged as given the broad categories used for the original shading (<90% conserved. >90% conserved), there were no actual changes to the image as pictured following reanalysis.

14) Figure 2: None of the axis labelling is legible

We believe this is now corrected.

Reviewer #2 (Remarks to the Author):

In the manuscript titled” Cross-lineage protection by vaccine-elicited human antibodies binding the influenza B hemagglutinin”, the authors used IBV HA probes in antigen-specific FACS sorting to interrogate humoral responses to IBV in humans. The authors found a significant proportion of IBV HA-specific B cells could recognize both Yamagata and Victoria lineages. A large panel of recovered mAbs were cloned and tested in ELISA, HIA and FRA experiments. This study provides clear conclusions and significant amounts of information and experimental data of IBV-reactive mAbs.

1. The labels in Figure 2 are unreadable.

We believe this is now corrected.

2. The authors generated the IBV stem proteins based on previous work for IAV (Yassine et al. 2015 Nature Medicine). HA stems are notoriously unstable and it has been known that stable trimeric HA stems are difficult to produce. It is possible that, in figure 3, the mAbs showed no binding activities to HA1s and stems are actually stem-directed. To prove the quality of the HA stems used in this study, it is recommended and Important that the authors include a diagram of their stem protein constructs and a SDS-PAGE gel (or size exclusion chromatography) showing the stable trimeric IBV stem proteins in supplementary section, which will add significant value to this manuscript.

A similar point regarding the IBV stem proteins was made by Reviewer 1, point 1. We agree that producing stable stem constructs is highly challenging and it is certainly possible that mAbs in Figure 3 (notably W85-3E10 and R95-1H08) which bind HA but not HA1 may also be stem-specific. We now include Supplementary Figure 12, which includes details of the sequences, expression characteristics and antigenic characterisation of the IBV stem probes used in this study as suggested. The prototypic stem mAb CR9114 fails to bind our constructs, which suggests antigenic differences compared to full-length HA. However, we are confident we have identified at least three novel stem mAbs based on a totality of binding information including binding to these preliminary stem constructs. It should be noted that the generation of IBV stem proteins is not the major focus of the current study, that this was the first attempt to produce them we have seen reported in the literature, and that a lack of available IBV-specific stem antibodies make the conformational assessment of these constructs difficult. As new IBV stem antibodies are published, this will facilitate improved IBV stem designs for future studies.

We have expanded the original text in the relevant results section:

“Since many broadly reactive IAV-specific mAbs bind the HA stem, we generated recombinant “stabilised” IBV stem constructs (Supplementary Figure 10) based upon the designs employed to generate the IAV stem domain³³. The recombinant IBV stem proteins failed to bind the prototypic stem mAb CR9114, indicating antigenic changes compared to full length HA (Supplementary Figure 10). However, the IBV stem proteins were bound by three putative broadly cross-reactive IBV stem-specific mAbs - W85-3F06, R95-1E03, R95-2A08. Stem specificity was further supported based upon binding to full-length HA but not to purified HA1 proteins by ELISA (Fig. 3).”

3. Subjects information was not clearly described in the manuscript. Were the three IIV4 recipients used for FACS sorting and mAbs isolation K77, W85 and R95? Also, including a demographic table summarizing the ethnic and age background of all the participants in this study in supplementary section would be informative.

We now provide a table with the age, gender and self-reported influenza vaccination history of all participants, and the subjects for mAb isolation (K77, W85 and R95) in Supplementary Figure 14. Ethnicity information was not collected in this study, however being based at the University of Melbourne, our cohorts draw heavily from the ethnically diverse student body.

4. It is interesting that the subjects (supplementary fig 4.) generated CR-E and CR-Y but no CR-V mB cells after immunization. The baseline mB cells of these subjects are also more Yamagata lineage dominant. It would be interesting to know the influenza exposure history (circulating IBV lineage in childhood, vaccination and previous infection record) of these subjects.

Correlating the extent of cross-reactivity within memory B cell populations with the previous exposures to IBV would be of tremendous interest. Unfortunately, within the cohorts in the current study (and indeed most studies) an accurate and reliable history of influenza exposure is not available. However, we now include information on seasonal vaccine administrations within the cohorts studied (self-reported, Supplementary Figure 14). The methods have been modified with the following addition:

“Participant information in summarised in Supplementary Figure 14.”

5. In the flow cytometry experiments, the authors observed the immune dominance to Yamagata lineage (13/Phuket) in the subjects. This might be an artifact due to the quality (stability, folding, quaternary structures or accessibility to mB cells) of the HA proteins (13PH and 08BR) used. To exclude this possibility, the authors might test other HA proteins (i.e. 06FL and 04MA, etc.) with a few representative subjects in the flow cytometry experiments.

We have stained some representative subjects with B/Florida/4/2006 and B/Malaysia/2506/2004 probes as suggested. The staining patterns were broadly consistent with our previous observation and further support that seasonal influenza vaccines drive a major expansion of cross-reactive memory B cells that display preferential binding to B/Yamagata vs B/Victoria probes. These data are included as new Supplementary Figure 6. We cannot exclude the possibility that qualitative aspects of each IBV probe might influence the extent and nature of binding to IBV-specific memory B cells. However, we do note that the B/PHU and B/BRIS probes did allow the isolation of 47 mAbs that bound a wide range of distinct epitopes (and the sequencing of many more antibody lineages), suggesting that no obvious antigenic defects were apparent in these novel reagents. Supporting this, we find the integrity of both B/PHU and B/BRIS HA probes appear equivalent as assessed via SDS-PAGE and/or gel filtration (Supplementary Figure 10).

The following text has been added to the Results:

“The extent to which preferential binding of cross-reactive memory B cells to the B/Yamagata probes is generalisable to all IBV strains is not clear. However, this observation was recapitulated in representative donors using historical B/Florida/4/2006 (B/Yamagata lineage) and B/Malaysia/2506/2004 (B/Victoria lineage) probes for IBV strains not included in the IIV3 or IIV4 vaccines (Supplementary Fig. 6).”

We have included the flow cytometry data as new Supplementary Figure 6:

Supplementary Figure 6. B/Florida and B/Malaysia cross-reactive B cell populations in four representative subjects receiving IIV4

Cross-reactive staining patterns in cryopreserved PBMC samples from four subjects taken 4 weeks post-IIV4 immunisation were assessed using recombinant HA probes derived from B/Florida/4/2006 and B/Malaysia/2506/04. Shown in comparison to samples previously stained with B/BRIS and B/PHU probes.

Alternatively, an indirect experiment can be performed to further confirm the observation: The authors might perform ELISA to test the subjects' serum antibody titers to HA proteins from different Yamagata and Victoria strains.

As suggested, we also performed a comprehensive series of ELISAs to examine serum reactivity before and after immunisation to a range of historical and recent IBV strains from both lineages. At baseline, we see a range of serological reactivity against each IBV lineage across individuals, with no consistent dominance of antibody responses that bound Yamagata lineages versus Victoria lineages. The extent of serum reactivity to each lineage was somewhat variable and likely heavily influenced by recent IBV exposure and vaccination histories within our cohorts.

Rebuttal Figure 1 – IBV HA-specific serum antibody responses at baseline

Serum endpoint titres of antibody binding the indicated IBV strains at baseline within subjects from both IIV3 and IIV4 immunisation trials (N=48 in total).

Of interest, we observed a consistently higher expansion in serum endpoint titres against all strains induced by IIV4 versus IIV3 immunisation, a finding consistent with both the inclusion of each IBV lineage and the preferential expansion of cross-reactive B cell populations observed by flow cytometry. We feel this data is worth including in the revised manuscript. However, delineation of the extent of truly cross-lineage serum antibody responses following these two immunisation regimens (IIV3 vs. IIV4) requires further exploration.

The following text has been added to the Results:

“The serological implications of cross-reactive B cell expansion are currently unclear. However, we observed consistently greater expansion in serum endpoint titres against diverse IBV strains in subjects receiving IIV4 compared to subjects receiving IIV3 (Supplementary Fig. 7), suggesting antibody binding both lineages might be elicited as part of a broad anti-IBV polyclonal response.”

We have included the serum ELISA data as new Supplementary Figure 7:

Supplementary Figure 7 – IBV HA-specific serum antibody responses following immunisation with seasonal influenza vaccines

The fold change in serum endpoint titres of antibody binding the indicated IBV strains following IIV3 and IIV4 immunisation was determined by ELISA. Median and IQR are indicated.

6. The authors stated that “out of 672 sorted B cells, 519 productive heavy sequences were recovered.” Please include the number of clones with both heavy chain and light chain sequences recovered.

The text has been modified to read:

“Out of 672 sorted B cells, 519 productive heavy sequences were recovered, 303 of which also had productive light chain sequences.”

7. Data in supplementary table 1. is fragmented and unreadable. Also, IGHV information is missing. Using an excel file to document these data might be better than using a PDF file.

We apologise for the poor clarity of the rendered pdf. Supplementary Table 1 is currently formatted as an excel file comprising relevant IGHV, IGKV and IGLV information.

8. A significant number of IBV reactive Ab clones were identified in this study. It might be worth trying to perform statistical analyses on the germline gene usage of the clones found in this study to identify “molecular signatures” of IBV specific antibodies. (Ref: Avnir et al., 2014 PLoS Pathogens & Joyce et al. 2016 Cell)

We have examined our sequence dataset for any signatures of IBV specific antibodies. However, we have not identified to date any clear patterns associated with IBV recognition in this case. The previous reports cited by the reviewer detail IGHV1-69 derived Group1-reactive antibodies (Avnir et al., 2014) or the stereotypic “classes” of immunoglobulins defined in multiple donors with Group1/Group2 IAV cross-reactivity (Joyce et al., 2016, Andrews et al., 2017). Notably, both of these examples bind highly similar epitopes restricted to the highly conserved stem domain of IAV HA. While the ~500 sequences we recovered in the current study seem significant, any convergence within the diverse polyclonal responses expected to target the entire HA protein likely require greater sequencing depth (or NGS approaches) and more study subjects than the current study provides.

Reviewer #3 (Remarks to the Author):

In this article by Liu et al., the authors analyze memory B cell (MBC) responses that are directed against influenza B virus HA glycoprotein. The authors use a flow cytometry-based approach to identify and sort HA-specific memory B cells that are specific to Phuket HA (Yamagata lineage) or Brisbane HA (Victoria lineage). The main time point where most of the analysis was performed was 4 weeks after immunization of healthy young adults with the trivalent (contains Yamagata, n=30) or the quadrivalent (contains Yamagata + Victoria, n=20) inactivated seasonal influenza. The authors elegantly show the distinct MBC populations directed against the Phuket HA vs. the Brisbane HA and those that equally bound to both probes. The authors further dissected the response by generating recombinant human mAbs from sorted single HA+ cells and characterizing the binding and functional capacity of the mAbs, confirming the mono-specificity or cross-reactivity of the sorted cells. Some of the mAbs were then shown to be broadly protective in a lethal murine influenza B virus challenge model. Further, the authors explored the role of the Fc-mediated effector functions in the protective capacity of a subset of the mAbs. Finally, the author generated and sequenced viral escape mutants against a subset of the in vitro neutralizing mAbs.

Overall the manuscript is very well-written with clear figures and the study set up is robust given the size of the cohorts. The flow cytometry-based approach that the authors used is innovative.

1) In the title, it is inaccurately stated that the analyzed MBCs/mAb are vaccine elicited. While this could be true for a part of the analyzed response, but given the sorting strategy

there is no definitive way to tell how much of the sorted MBCs are actually vaccine-induced vs. pre-existing. This is supported by data shown in Fig. 1D, where only a little over 15% of Phuket HA+ MBCs detected at day 28 after vaccination were CD21lo, a sign of recent activation. To truly and properly study vaccine-elicited responses, the authors should have analyzed the d7 plasmablast responses as described in detail by numerous reports over the last 10 years. Recently generated plasmablasts are short lived in blood so no interference from pre-existing cells would exist. Contrary to the perpetuated myth, plasmablasts can be analyzed from cryo-preserved PBMCs especially using flow cytometry.

B cell responses to seasonal vaccination in adults have been reported to draw heavily from the memory B cell pool (Lee et al., 2016, Jiang et al., 2013 and others). Plasmablast studies, as mentioned by the reviewer, are indeed done by a number of groups and are particularly useful for clarifying the recruitment of given B cell clones into the serological response. However, our complementary approach to study human vaccine responses based on rHA probes allows the assessment of specificity and in particular, cross-reactivity, within the MBC pool of a large number of clinical donors.

We do agree that interrogating plasmablast responses to clarify the degree to which IBV cross-reactive B cell populations are activated early and recruited to the bone marrow is an important question we aim to investigate in future studies. In the current study, we did not collect D7 PBMC samples (when plasmablast responses peak) from seasonal vaccination cohorts, which we instead focussed on MBC responses that peak later. An additional complication is that plasmablasts do not bind rHA probes efficiently or at all (with the exception of some IgA expressing plasmablast subsets) due to rapid downregulation of surface B-cell receptor expression following activation (Koutsakos et al., 2018). We cannot therefore employ an analogous approach to measure plasmablast specificity as we do with MBC. Furthermore, this approach would require the generation of impractically large libraries of expressed immunoglobulins for extensive *in vitro* testing.

In terms of being “vaccine-elicited”, we believe that IBV rHA+ B cell populations as reported in this study (Fig1, Fig2, Sup Fig4) clearly undergo a polyclonal expansion in response to immunisation, consistent with our observations from previous human influenza vaccine trials (Wheatley et al., 2015, Joyce et al., 2016, Wheatley et al., 2016, Andrews et al., 2017, Koutsakos et al., 2018). While not all were CD21lo (recently activated), a significant proportion were and expansion in frequency can clearly be seen across both lineage-specific and cross-reactive B cell populations in Supplementary Figure 4. Therefore, while the B cell populations studied demonstrate vaccine responsiveness, we see the reviewer’s point that we cannot categorically state that each specific B cell lineage from which we isolate mAbs was vaccine “elicited”. We have removed “vaccine-elicited” from the title as suggested.

Old title:

Cross-lineage protection by vaccine-elicited human antibodies binding the influenza B hemagglutinin

New title:

Cross-lineage protection by human antibodies binding the influenza B hemagglutinin

2) The HA staining looks robust, but there is no negative control. An irrelevant, similarly labeled probe should have been used to be able to accurately tell whether all these HA+ MBCs are truly specific. This is especially true for the pre-vaccination timepoint. Without that, the data in Fig. 1B could turn out to be an overestimate.

While use of an irrelevant similarly labelled probe is possible, this actually does not give an accurate indication of background for the HA probes, as these will identify specific and poly-reactive B cells that bind the irrelevant antigen, likely different cells and at different frequencies for any given antigen. In the current study, to reduce non-specific staining in our analysed clinical samples, we do include irrelevant streptavidin-BV510 conjugate and dump all BV510+ B cells prior to analysing the HA probes (SA-PE, SA-APC or SA-AX488). While a degree of residual non-specific staining (background) might be expected with the use of rHA probes, in our experience this is low and any resultant overestimation of MBC frequencies is also likely to be very small. Specificity of the rHA probes has been comprehensively established in the current and a number of previous studies using several complementary approaches including:

- (i) use of both FMOs and irrelevant probe controls (Whittle et al., 2014, Wheatley et al., 2015)
- (ii) blocking rHA probe binding using sheep anti-HA sera (Figure S4, Koutsakos et al., 2018)
- (iii) comparison of rHA staining frequencies with ELISPOT (Figure S8, Koutsakos et al., 2018)
- (iv) clonal analysis of the BCR sequencing (Supplementary Figure 8) indicates the presence of numerous clonal expansions. Clonal expansions are a reliable indicator of antigenic specificity, as it is unlikely that two or more non-specific B cells of the same clone will bind the probes by chance. Across the four populations, we found over 65% of BCR sequences recovered were from clonally expanded lineages, with a natural tail of singletons. As clonality is a function of sequencing depth, it is likely that many singletons are also HA-specific, as evidenced in (iv).
- (v) the efficient recovery of 46 mAbs from sorted IBV probe+ cells in the current paper conclusively demonstrates that the rHA probe binding is highly specific.

3) The correlation in Fig. 1E is non-sensical as it has been shown multiple times that the increase in serum Abs is after influenza vaccination correlates with the frequency of the antibody-secreting plasmablasts.

We concur that the correlation between plasmablasts and serum antibody has been well established. However, the correlation between MBC and serum antibody for IBV has not been established. Figure 1E illustrates that the expansion in the frequency of IBV-specific memory B cells in response to immunisation also correlates with the increase in titres of anti-IBV serum antibodies. We previously reported a similar observation for H5N1 (Wheatley et al., 2015) and H1N1 and H3N2 (Wheatley et al., 2016).

4) The authors show the binding of their HA probe to murine GC B cells for no apparent reason.

Infected mice provided a platform to validate that the recombinant IBV HA probes can bind B cells elicited by the wild-type IBV virus *in vivo*. The relevant text has been modified to make this clearer and now reads:

“The antigenic conservation of the B-PH13 HA probe as compared to the wild-type virus was further confirmed by staining B cells in mice experimentally infected mice with B/Phuket/3073/2013 (Supplementary Fig. 3).”

5) There are key questions/findings that the authors could have addressed/highlighted, such as the extent to which B cell responses in humans that are directed against the Yamagata lineage vs. Victoria lineages vs. cross-reactive ones are elicited following seasonal vaccination. Do we actually need a quadrivalent vaccine or the trivalent vaccine is enough? Are the data showing that the majority of the cross-reactive mAbs are non-neutralizing compared to strain-specific ones statistically significant? If so, what are the implications? The authors instead were seemingly side-tracked with a population of HA-binding cells that did not express any mAbs with any particularly special capacities. And even that population, the authors did not provide a meaningful explanation for it.

We concur that influenza B, particularly human immunity to influenza B, remains chronically understudied. Generalisable statements about the current seasonal vaccines, such as the relative merits of trivalent versus quadrivalent immunisation are at this point difficult to distil from our studies and others to date. However, we believe our study has now established a solid platform of reagents and techniques that will both significantly contribute to the field and serve as a springboard allowing us and others to narrow some of the key knowledge gaps identified by the reviewer. We have expanded the discussion with the following paragraph:

“A limitation of our study is the restricted number of mAbs and the heterogenous patterns of IBV HA binding and neutralisation. However, in general we find seasonal influenza vaccines drive expansion of two predominant populations of memory B cells; one binding neutralising epitopes proximal to the receptor binding site shared within each respective IBV lineage, and a second population of highly cross-reactive B cells binding both lineages and expressing antibodies that were generally non-neutralising. Although many human mAbs displayed a degree of protection in mice, the protective benefit of cross-reactive non-neutralising antibodies in human populations remains an open question. Nevertheless, IIV4 drove expanded serological reactivity against a broad range of IBV strains and simultaneously expands B cells expressing HIA+ antibody with pan-B/Yamagata or pan-B/Victoria reactivity, suggesting IIV4 has an improved capacity to combat antigenic drift in either lineage while removing the risk of lineage mismatch relative to IIV3.”

Reviewers' comments:

Reviewer #1 (Remarks to the Author):

1) The stem construct should not be called 'stabilised' and it should be clearly called out in the main text that is not folded correctly.

2) 5A7 is not a pure anti-stalk antibody but binds to the head-stalk interface. It actually makes all its contacts with HA1. W85-3F06 on the other side does not bind HA1 at all and still exhibits HI activity. This suggests a mix up or a technical issue with the assay(s). The assays should be repeated. Also, would W85-3F06 be still HI active if RDE treated?

Reviewer #2 (Remarks to the Author):

In the manuscript "Cross-lineage protection by human antibodies binding the influenza B hemagglutinin", the authors used an innovative approach to interrogate humoral responses to IBV. This is a comprehensive study that provides significant amounts of data of the understudied IBV with clear conclusions. The authors have extensively revised the manuscript. All my concerns have been addressed.

Reviewer #3 (Remarks to the Author):

The authors have adequately addressed my concerns.

RESPONSES TO REVIEWERS' COMMENTS

We thank the reviewers and address each reviewers' comments below in point-by-point form. We appreciate the opportunity to re-submit our study to *Nature Communications*.

Reviewer #1 (Remarks to the Author):

1) The stem construct should not be called 'stabilised' and it should be clearly called out in the main text that is not folded correctly.

We have removed any references to stabilised. The main text now indicates the lack of correct folding. This section now reads:

“Since many broadly reactive IAV-specific mAbs bind the HA stem, we generated recombinant IBV stem constructs (Supplementary Figure 10) based upon the designs employed to generate the IAV stem domain³³. These preliminary recombinant IBV stem proteins appeared misfolded and failed to bind the prototypic stem mAb CR9114, indicating antigenic changes compared to full length HA (Supplementary Figure 10). However, the IBV stem proteins were bound by three putative broadly cross-reactive IBV stem-specific mAbs - W85-3F06, R95-1E03, R95-2A08. Stem specificity was further supported based upon binding to full-length HA but not to purified HA1 proteins by ELISA (Fig. 3).

2) 5A7 is not a pure anti-stalk antibody but binds to the head-stalk interface. It actually makes all its contacts with HA1. W85-3F06 on the other side does not bind HA1 at all and still exhibits HI activity. This suggests a mix up or a technical issue with the assay(s). The assays should be repeated. Also, would W85-3F06 be still HI active if RDE treated?

The previously reported HI activity for W85-3F06 was very weak (HAI dilution 10), at the detection limit of the assay. The mAb W85-3F06 was re-expressed and HI assays repeated, with W85-3F06 mediating no HI activity. Notably, all controls, other human mAbs and W85-3F06 displayed activity within a single dilution of the previous assay, confirming their wider comparability (shown below).

Haemagglutination Inhibition Assay - WHO Influenza Centre, Melbourne						
		Repeated Assay			Initial Assay	
		Virus	B/BRISBANE/60/2008	B/LEE/40	B/BRISBANE/60/2008	B/LEE/40
		Passage #	E7	X,E3	E7	X,E3
		Animal Number	Turkey #48	Turkey #48	Turkey #31	Turkey #31
Sample ID						
BR08-specific	K77-2E02		1280	<10	640	20
	R95-1E12		5120	1280	5120	1280
PH13-specific	K77-1G12		<10	2560	10	1280
	W85-1B01		<10	10240	<10	5120
Cross-reactive, non-stem	R95-1C01		<10	<10	<10	<10
	R95-1E05		<10	<10	<10	<10
	R95-1F04		<10	<10	<10	<10
	R95-1E03		<10	<10	<10	<10
Cross-reactive, "stem"	R95-2A08		<10	<10	<10	<10
	W85-3F06		<10	<10	10	10
	W85-3F06* (RDE treated)		<10	<10		
Controls	F2425 B/BRISBANE/60/2008 ferret antisera		5120	20	5120	40
	Neg Control D0 Ferret		<10	<10	<10	<10
	PBS		<10	<10	<10	<10

On balance, we agree that W85-3F06 does not have HI activity, consistent with the other two stem mAbs (R95-1E03 and R95-2A08) identified in the study which display similar binding properties to W85-3F06. We have updated the relevant boxes in Figure 3 to indicate W85-3F06 has no HI activity. The main text has been updated to remove any reference to HI activity from W85-3F06.

Reviewer #2 (Remarks to the Author):

In the manuscript “Cross-lineage protection by human antibodies binding the influenza B hemagglutinin”, the authors used an innovative approach to interrogate humoral responses to IBV. This is a comprehensive study that provides significant amounts of data of the understudied IBV with clear conclusions. The authors have extensively revised the manuscript. All my concerns have been addressed.

Reviewer #3 (Remarks to the Author):

The authors have adequately addressed my concerns.

REVIEWERS' COMMENTS:

Reviewer #1 (Remarks to the Author):

All issues have been addressed. Nice paper.